# Source reconstruction via deposition measurements of an undeclared radiological atmospheric release

Stijn Van Leuven[1,2,3], Pieter De Meutter[1,2], Johan Camps[1], Piet Termonia[2,3], and Andy Delcloo[2,3]

[1]Belgian Nuclear Research Centre, Mol, Belgium
[2]Royal Meteorological Institute of Belgium, Brussels, Belgium
[3]Department of Physics and Astronomy, Ghent University, Ghent, Belgium

**Correspondence:** Stijn Van Leuven (stijn.van.leuven@sckcen.be)

**Abstract.** Inverse modelling of atmospheric releases of radioactivity consists of reconstructing the release source by combining radiological field measurements with atmospheric transport calculations. This is typically performed with air concentration measurements, although deposition measurements or gamma dose rate measurements could also be used. In this paper, we assess the use of deposition measurements of radioactivity in this context. This is done through a case study of the undisclosed release of the radionuclide $^{106}$Ru in Eurasia during the autumn of 2017. The atmospheric transport model we utilise for this purpose is Flexpart. Inverse modelling is performed with the inverse modelling tool FREAR, which has been modified to work with deposition measurements. The inversion consists of Bayesian and cost function based algorithms to reconstruct the initial source properties. Inverse modelling is applied to both real and synthetic deposition data following the $^{106}$Ru release. We also construct synthetic air concentration data for use in inverse modelling, to make a comparison with the results using deposition data. It is found that source localisation is feasible with both the synthetic and real world deposition data. Synthetic air concentration measurements lead to more precise source localisation than deposition. It is demonstrated that this can be explained by the lower detection limits of air concentration measurements compared to deposition.

## 1 Introduction

The ability to reconstruct the sources of polluting atmospheric releases is critical in the endeavor of monitoring and guarding the health of man and nature. The process of such source reconstruction, also often referred to as inverse atmospheric transport modelling (or simply 'inverse modelling'), has been applied to the releases of pollutants such as greenhouse gases (Stohl et al., 2009; Houweling et al., 2015; Henne et al., 2016), volcanic sulphur dioxide (Eckhardt et al., 2008; Kristiansen et al., 2010), microplastics (Evangeliou et al., 2022), radionuclides (Devell et al., 1995; De Cort, 1998; Davoine and Bocquet, 2007; Stohl et al., 2012; Katata et al., 2015; De Meutter and Hoffman, 2020) and others.

Specifically, the release of radionuclides can potentially pose an immediate danger to the surrounding population, due to either direct exposure to radiation from airborne or deposited radionuclides, or from contaminated water and foodstuffs. Given

these hazards, knowledge of the source term is crucial for emergency preparedness and response. Another relevant application of inverse modelling of radiological releases is the verification of the Comprehensive Nuclear-Test-Ban Treaty (CTBT). The CTBT has been adopted by the United Nations in 1996 – though yet to be ratified by all Annex II states – to ban all nuclear explosions. Adherence to the CTBT is monitored by, among other methods, almost 80 radionuclide detection stations worldwide. In order to link a CTBT-relevant waveform event (as can be identified by seismic, ultrasound or hydro-acoustic signals) to a nuclear explosion, these radionuclide measurements can be used to reconstruct the source with inverse modelling techniques (Wotawa et al., 2003; Eslinger and Milbrath, 2024; De Meutter et al., 2024).

Following detections of radionuclides, one can use various inverse modelling techniques to reconstruct the source term. Combining observations and results from an atmospheric transport modelling (ATM) in a mathematically consistent manner makes it possible to estimate properties of the source, such as its location, the total amount of material released and the timing of the release.

The most commonly used quantity in inverse modelling in the radiological context is activity air concentration. However, also often available are measurements of dry and/or wet deposition. Such measurements thus do not contain noble gases (such as xenon), since these nuclides are not subject to deposition (Nebeker et al., 1971; Slinn, 1984). Some common nuclear fission products that have been previously detected in deposition samples after a radiological release are $^{137}$Cs, $^{134}$Cs, $^{131}$I and $^{106}$Ru (Evangeliou et al., 2017; MEXT (Ministry of Education, Culture, Sports, Science and Technology), 2011; Ramebäck et al., 2018; Masson et al., 2019).

In practice, deposition measurements can provide advantages compared to those of air concentration. Activity air concentration detectors are typically part of expensive, stationary networks. Deposition, on the other hand, can be obtained by mobile and cheaper deposition collection methods. For instance, following a known or suspected release, one can place deposition collectors in locations that are likely to be hit according to the current or forecast meteorological conditions. Moreover, in this way a plume that misses existing air concentration detectors can still be captured by placing deposition tanks, assuming the collected deposition surpasses the detection threshold. Deposition can also be collected a-posteriori, by taking soil and plant samples. However, for a given release, air concentration is generally more easily detected than deposition due to the lower detection limits of air concentration detectors.

Deposition detections of radionuclides have previously been used in combination with those of air concentration to estimate the source term of the Chernobyl (1986) and Fukushima (2011) nuclear disasters (Evangeliou et al., 2017; Stohl et al., 2012; Winiarek et al., 2014; Dumont Le Brazidec et al., 2023). In these cases the source location was already known beforehand, significantly simplifying the inverse modelling procedures by reducing the overdetermination of the problem. However, the source location is not always known. Such real-life scenarios include potential CTBT-relevant events and the undisclosed release of $^{106}$Ru during late September 2017.

During late September and early October of 2017, anomalous amounts of the radionuclide $^{106}$Ru ($T_{1/2} = 1$ y 7 d), and to a lesser extent $^{103}$Ru ($T_{1/2} = 39.3$ d), were detected in Europe and other parts of the northern hemisphere (Masson et al., 2019). To this day, no release of the radioactive ruthenium has been officially declared. The source term has been estimated in previous studies (Sorensen, 2018; Shershakov et al., 2019; Saunier et al., 2019; Western et al., 2020; Le Brazidec et al.,

2021; Tollose et al., 2021). The results of these studies are summarised in Table 1. Most studies conclude that the release likely originated in the southern Ural region, with the Federal State Unitary Enterprise "Mayak Production Association" being the most probable source, as it is the only nuclear facility in the region. The released activity varies from some 100's of TBq to about 1 PBq, with the majority of the release between 24 and 26 September 2017. In these studies, only atmospheric concentration measurements of $^{106}$Ru, of which there are more than 1000 available (Masson et al., 2019), were used directly in the inverse modelling process. However, $^{106}$Ru was also detected in numerous deposition samples. Masson et al. (2019) has aggregated 135 deposition detections in total. Values up to $\sim 300$ Bq m$^{-2}$ in Russia and up to $\sim 90$ Bq m$^{-2}$ in Europe (Scandinavia) were detected.

The anomalous $^{106}$Ru release of 2017 serves as a valuable test case due to the absence of any prior radioactive ruthenium background. $^{106}$Ru does not occur naturally, and that from the only previous major release (the Chernobyl nuclear disaster in 1986) has long since decayed due to its approximately 1 year half-life. Thus, there is no background-related error associated with interpreting the $^{106}$Ru detections. This contrasts with, for example, $^{133}$Xe or $^{137}$Cs detections, which can contain traces from present-day civil sources (Gueibe et al., 2017) and historical nuclear accidents and weapon tests (De Cort, 1998; Evangeliou et al., 2016) respectively.

**Table 1.** Source term estimates in existing literature of the undeclared $^{106}$Ru release of 2017. The Mayak Production Association is located at 55.71° N, 60.85° E.

| Reference | Location | Total activity | Release date |
|---|---|---|---|
| Sorensen (2018) | Mayak | 1100 TBq | 26 Sep. |
| Shershakov et al. (2019) | South and Central Urals | 500 TBq | 25–26 Sep. |
| Saunier et al. (2019) | Mayak | 250 TBq | 26 Sep. |
| Western et al. (2020) | Mayak | 441 TBq | 24 Sep. |
| Le Brazidec et al. (2021) | [55°, 56°] N, [59°, 61°] E | 200–450 TBq | 25–26 Sep. |
| Tichý et al. (2021) | Mayak | 130–344 TBq | 25–26 Sep. |
| Tollose et al. (2021) | Mayak | 620 TBq | 23–26 Sep. |

In this paper, the source of the undisclosed $^{106}$Ru release is reconstructed based on the available deposition detections using the FREAR inverse modelling code (De Meutter et al., 2018; De Meutter and Hoffman, 2020; De Meutter et al., 2024). The source location is assumed unknown for the purposes of inverse modelling. The intent of this paper is not necessarily to refine the source term parameters of the $^{106}$Ru release, but rather to evaluate the capabilities of inverse modelling with deposition measurements.

The rest of the paper is organised as follows. In Sect. 2, we describe the measurement data and models used for this study, as well as the constructed inverse modelling experiments. In Sect. 3, the results are shown and discussed after the data and model setup are evaluated. The conclusions are contained in Sect. 4.

## 2   Data, models and experiments

Described herein are the atmospheric transport model, the meteorological data, and the inverse modelling algorithms. We also include a sub-section describing the inherent physical differences between measurements of air concentration and deposition, as is relevant for inverse modelling. The section also contains a description of the different experiments.

### 2.1   $^{106}$Ru deposition data

Deposition data of $^{106}$Ru located across the Eurasian continent following the 2017 anomalous release has been compiled by Masson et al. (2019). The dataset contains 135 deposition measurements in total. For this study, we have made a selection of data points for use in the inverse modelling calculations based on their temporal measurement window, location and the physical quantity that was measured. Only observations that started after 2 September 2017 and ended before 25 October 2017 were selected. After one month the released material has dispersed significantly in the atmosphere so that it generally no longer contains relevant information for the purpose of inverse modelling. Furthermore, only detections at a distance greater than 1000 km from Mayak were selected. Detections closer than this may be confounded by particle-gaseous partitioning of the radioactive ruthenium and local weather effects. The absence of gaseous $^{106}$Ru has only been confirmed in Europe (Masson et al., 2019). Finally, only detections of activity per surface area (i.e. Bq m$^{-2}$) were selected. There were five measurements in the remaining dataset reported in activity per precipitated volume (i.e. Bq L$^{-1}$), which were not used. Values in Bq m$^{-2}$ are directly suitable for the inverse modelling framework, as the deposition values of the ATM we use is output in these units. Thus, to keep the dataset consistent, we decided to use the Bq m$^{-2}$ values.

The end result of applying all the above criteria leaves 30 remaining measurements. We follow the distinction made by Masson et al. (2019) to label 18 of these "activity concentration in rain water" and 12 "dry + wet fallout", which we will abbreviate to "rain water" and "fallout" respectively. The locations of these 30 remaining measurements are shown in Figure 1. Though the description "activity concentration in rain water" may seem to imply only the collection of wet deposition, in general monitoring networks do not discriminate between dry and wet deposition. It is therefore assumed that both the rain water and fallout measurements contain dry and wet deposition collected over the entire measurement window. We will perform the inverse modelling separately with the rain water and fallout datasets, as well as by combining both datasets.

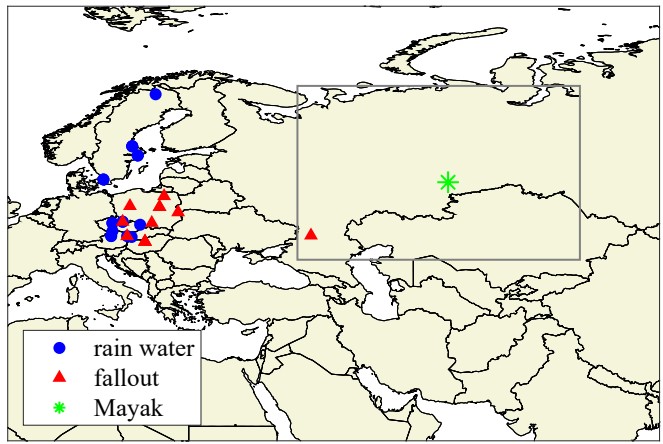

**Figure 1.** Location of 18 rain water (blue circles) and 12 fallout (red triangles) deposition observations selected from the supplementary material in Masson 2019. Some measurement locations (partially) overlap. The Mayak nuclear installation is located at the green star. The grey rectangle defines the search area for the inverse modelling calculations (lower-left corner $[40°\,\mathrm{E}, 45°\,\mathrm{N}]$ and upper-right corner $[80°\,\mathrm{E}, 70°\,\mathrm{N}]$).

One remaining aspect is the timing to be used for the measurements. The duration of the measurement windows range from
1 day (13 measurements) to 7 days (8 measurements), with the rest scattered in between (8 measurements), except one which
is 28 days. However, the deposition data in the supplementary material of Masson et al. (2019) only provide the start and end
dates of the measurements. Absent are the hours at which the measurements may have started or ended. Assigning start and end
hours is however necessary to perform the inverse modelling, lest it is assumed that each measurement started and ended at 0:00
UTC, which does not appear realistic. The choice made for this study is that each measurement starts at 5:00 UTC and ends
at 13:00 UTC. It is reasonable to assume the start and end times to be similar. However, the choice to extend the measurement
interval by 8 h was made to increase the likelihood of capturing the relevant precipitation event that contributed to any wet
deposition. Whilst a-priori this may increase the chance of erroneously capturing a rain event outside the true measurement
interval, we show in Sect. 3.1 this is not the case. The time extension on the measurement window is kept more modest than the
theoretical maximal error, since such a large window would induce chances of capturing erroneous precipitation and (retro-)
plume dispersion.

## 2.2 Atmospheric transport modelling

The source-receptor sensitivities (SRS) were calculated with the stochastic Lagrangian particle dispersion model Flexpart
v10.4 (Stohl et al., 2005; Pisso et al., 2019), used in backwards-in-time mode (Seibert and Frank, 2004; Eckhardt et al., 2017).
The backward-in-time mode is based on the adjoint version of the ATM, which mathematically equates to simply inverting
the sign of the advection term in the transport equation for a Lagrangian particle model (Thomson, 1987; Flesch et al., 1995;
Pudykiewicz, 1998). SRS fields can be obtained through both forward- and backward-in-time calculations. The forward- or
backward-in-time method will be more computationally efficient if the amount of observations is respectively greater or less

than the amount of potential geotemporal source term segments. In this study the source location is assumed unknown for the purposes of inverse modelling, therefore the backwards-in-time method is more efficient. Ten million particles were released for each backward-in-time calculation. The SRS fields were output every 3 hours on a 0.5° by 0.5° grid that covers the grey rectangle of Fig.1. The retro-plume dispersion was calculated from the end of each measurement, backward-in-time to 0:00 UTC 22 September 2017 which is several days before the release starts in existing literature (Table 1).

The relevant [106]Ru deposition parameters used in the Flexpart simulations are given in Table 2. The wet deposition parameters ($C_{rain}$, $C_{snow}$, $CCN_{eff}$ and $IN_{eff}$) are taken from Van Leuven et al. (2023) who found that the default deposition parameters were cause for an underestimation of global [137]Cs concentration following the Fukushima nuclear accident when using Flexpart v10.4. The in-cloud scavenging efficiencies are greater than one, but can mathematically be absorbed into other internal parameters of Flexpart (such as the cloud water replenishing rate), thus not necessarily violating the physical correctness of the model. Masson et al. (2019) found that the particle sizes were in the sub-micron range, hence the default value of 0.6 μm was kept.

| parameter name | symbol | value |
|---|---|---|
| rain scavenging coefficient | $C_{rain}$ | 3.6 |
| snow scavenging coefficient | $C_{snow}$ | 1.4 |
| cloud condensation nucleation efficiency | $CCN_{eff}$ | 1.8 |
| ice nucleation efficiency | $IN_{eff}$ | 1.6 |
| average particle diameter | $\overline{d}$ | 0.6 μm |
| particle diameter geometric variance | $\sigma_d$ | 0.3 |
| particle density | $\rho$ | 2500 kg m$^{-3}$ |

**Table 2.** Deposition parameters used for the aerosolised [106]Ru in the Flexpart simulations.

The input numerical weather data for the ATM calculations were obtained from the MARS archive from ECMWF with the use of the FlexExtract v7 software (Tipka et al., 2020). We evaluate two sets of meteorological data with different resolutions, both extracted from the same underlying ECMWF data. One is a nested set of hourly model values. This data consists of analyses at 0, 6, 12 and 18Z, intermixed with short forecasts of +1, +2, +3, +4 and +5h. The nesting is as follows: a 0.1° by 0.1° grid with the coverage of the full extent of Fig. 1 (lower-left corner [0°E, 20°N] and upper-right corner [90°E, 80°N]) is nested in a grid of 0.5° by 0.5° covering the northern hemisphere. The other set of meteorological data we use is a northern hemisphere model with 1° horizontal resolution and short forecasts of +3 h, resulting in a temporal resolution of 3 h. Both sets of meteorological data contain 137 non-uniformly spaced hybrid vertical levels ranging from 10 m above the surface up to approximately 80 km. The high resolution nested model will be denoted by (0.1°, 1 h) and the lower resolution model by (1°, 3 h).

## 2.3 Inverse modelling

The general idea behind inverse modelling is to estimate relevant source term parameters, given a set of observations and source-receptor sensitivities obtained by atmospheric transport modelling. Inverse modelling is most straightforward with a linear atmospheric transport model, such as Flexpart. A linear ATM has physical quantities $\phi_i$ (e.g. air concentration, deposition, etc.) that scale linearly with the geotemporal release segment $S_j$ of the source term:

$$\phi_i = \sum_j M_{ij} S_j. \tag{1}$$

The proportionality factors $M_{ij}$ are the source-receptor sensitivities (SRS) and form the components of the SRS matrix $M$. The SRS value $M_{ij}$ captures the sensitivity of observation $i$ to the geotemporal release segment $j$. Under this formalism, one needs to calculate the SRS values $M_{ij}$ only once in order to be able to generate $\phi_i$ for a given scaling of $S_j$.

The inverse modelling for this study is performed using the inverse modelling code FREAR (De Meutter et al., 2018; De Meutter and Hoffman, 2020; De Meutter et al., 2024). FREAR (Forensic Radionuclide Event Analysis and Reconstruction) is an open-source inverse modelling tool developed to aid nuclear emergency preparedness and response, and the CTBT verification regime. It takes as input a set of activity air concentration measurements and source-receptor sensitivities from ATM. Inverse modelling in FREAR can be performed with a Bayesian inference and a cost function optimisation method, as well as some other, more simple methods (a correlation and an overlapping retro-plume method).

The source term is assumed to be contained within the grey rectangle of Fig. 1, which is defined by a lower-left corner of $[40°\mathrm{E}, 45°\mathrm{N}]$ and upper-right corner $[80°\mathrm{E}, 70°\mathrm{N}]$.

### 2.3.1 Bayesian inference

The Bayesian method uses a Gaussian likelihood, where the standard deviation has been replaced by an inverse gamma distribution for the combined model and observation uncertainties (Yee, 2012):

$$p(\phi_{\mathrm{mod},i} | \phi_{\mathrm{obs},i}) = \frac{\overline{\alpha}^{\overline{\beta}} \Gamma(\overline{\beta} + 1/2)}{\sqrt{2\pi} s_i \Gamma(\overline{\beta})} \frac{1}{[\overline{\alpha} + (\phi_{\mathrm{obs},i} - \phi_{\mathrm{mod},i})^2 / (2s_i^2)]^{\overline{\beta}+1/2}}. \tag{2}$$

The hyperparameters $\overline{\alpha}$ and $\overline{\beta}$ of the inverse gamma distribution are fixed at $1/\pi$ and $1$, respectively. $s_i$ represents the combined model and observation uncertainties:

$$s_i^2 = \sigma_{\mathrm{obs},i}^2 + \sigma_{\mathrm{mod},i}^2. \tag{3}$$

Since the model errors are unknown but assumed to be dominant over the observational errors, the combined model and observation uncertainties are parameterised as

$$s_i = \sigma_{\text{mod},i} = \max(4\text{MDQ}_i, 0.5\phi_{\text{obs},i}), \tag{4}$$

where $\text{MDQ}_i$ is the minimal detectable quantity for observation $\phi_{\text{obs},i}$. The Bayesian algorithm takes into account detections, non-detections, misses and false alarms through the use of the detection limits (De Meutter and Hoffman, 2020). The posterior is sampled with the general-purpose Markov chain Monte Carlo algorithm MT-DREAM$_{\text{(ZS)}}$ (Laloy and Vrugt, 2012). In the Bayesian algorithm, the source term is parameterised by a singular block release specified by a longitude-latitude location, a start time, end time, and total amount released. The prior distribution for the start time is chosen uniform from 23 to 29 September 2017. The prior for the end time is determined by a uniform distribution on $r_{\text{end}}$ in

$$t_{\text{end}} = t_{\text{start}} + r_{\text{end}}(t_{\text{max}} - t_{\text{start}}). \tag{5}$$

This construction is used to ensure the end time occurs after the start time. The prior distributions of the source latitude, longitude and release quantity are also chosen uniform.

### 2.3.2 Cost function minimisation

The cost function method is based on minimising a modified version of the geometric variance (De Meutter et al., 2024):

$$F = \exp\left\{ \frac{1}{N} \sum_{i=1}^{N} \left[ \log(f(\phi_{\text{obs},i})) - \log(f(\phi_{\text{mod},i})) \right]^2 \right\}, \tag{6}$$

with

$$f(\phi) = \begin{cases} \dfrac{\phi^2}{4\text{MDQ}} + \text{MDQ} & \text{if } \phi \leq 2\text{MDQ} \\ \phi & \text{else}, \end{cases} \tag{7}$$

This allows the cost function algorithm to take into account detections and non-detections. The cost function $F$ does not optimise for the source location explicitly. Rather, the cost function optimisation is applied to each grid box by fixing the source location while varying the temporal release profile. The end result is a grid of residual costs, where lower cost means a better fit for a release at that location. The temporal release profile is parameterised as a sequence of block releases for each day, in contrast to the Bayesian method which only evaluates a single block release as part of the source term.

### 2.3.3 FREAR with deposition

The existing version of FREAR takes observation sets of activity air concentration as input. For this study, the FREAR source-code has been modified to work with both concentration and deposition observations. The source reconstruction can be performed using any combination of the different types of measurements simultaneously. Mathematically, this can be more clearly

denoted by writing Eq. (1) as $\phi = MS$ in vector notation, where $\phi$ and $S$ are column vectors. If $\phi$ contains multiple types of measurements, this can be represented as

$$
\begin{bmatrix} \phi_{\mathrm{conc}} \\ \phi_{\mathrm{tot}} \\ \phi_{\mathrm{wet}} \\ \phi_{\mathrm{dry}} \end{bmatrix} = \begin{bmatrix} M_{\mathrm{conc}} \\ M_{\mathrm{tot}} \\ M_{\mathrm{wet}} \\ M_{\mathrm{dry}} \end{bmatrix} S,
\tag{8}
$$

where the subscripts 'conc', 'tot', 'wet' and 'dry' stand for air concentration, total deposition, wet deposition and dry deposition, respectively. $M_{\mathrm{tot}}$ can not be calculated directly with Flexpart. Instead, the dry and wet components need to be calculated separately and added to obtain the total deposition:

$$
M_{\mathrm{tot}} = M_{\mathrm{dry}} + M_{\mathrm{wet}}.
\tag{9}
$$

### 2.4 Air concentration versus deposition

In this section we take the opportunity to expound on the physical differences between measurements of air concentration and deposition, as relevant for inverse modelling. These differences arise from the fact that the part of the concentration plume that is sampled with each type of measurement differs significantly.

Consider first that the source-receptor relationship (1) can also be written in its continuous form using inner product notation:

$$
\phi_i = \langle M_i, S \rangle = \int \mathrm{d}t \int_{\mathcal{D}} \mathrm{d}^3x \, M_i(x,y,z,t) S(x,y,z,t),
\tag{10}
$$

where $\mathcal{D}$ is the domain under consideration. Using the dual relation of the inner product, $\phi_i$ can also be related to the air concentration field $c(x,y,z,t)$ (Pudykiewicz, 1998; Yee et al., 2008):

$$
\phi_i = \langle c, \tilde{S}_i \rangle = \int \mathrm{d}t \int_{\mathcal{D}} \mathrm{d}^3x \, c(x,y,z,t) \tilde{S}_i(x,y,z,t).
\tag{11}
$$

Here $\tilde{S}_i$ is the source term for the adjoint transport equation of observation $i$, therefore referred to as the adjoint source function. $\tilde{S}$ takes a different form depending on the measured quantity:

$$
\tilde{S}_i(x,y,z,t) = \begin{cases} K_{\mathrm{conc},i}(x,y,z,t) & \text{for } \phi_{\mathrm{conc},i} \\ K_{\mathrm{wet},i}(x,y,t)\Lambda(x,y,z,t)T & \text{for } \phi_{\mathrm{wet},i} \\ K_{\mathrm{dry},i}(x,y,t)\dfrac{v_{\mathrm{dry}}(x,y,z,t)}{h}T & \text{for } \phi_{\mathrm{dry},i}. \end{cases}
\tag{12}
$$

$K_i$ functions as a receptor kernel that represents the geotemporal sampling space of the detector of observation $i$: each obser-
vation has a certain spatial coverage and covers a certain time period. For an air concentration measurement, this is a three
dimensional kernel with units $[L]^{-3}$ and the condition that $\int \mathrm{d}t \int_{\mathcal{D}} \mathrm{d}^3 x K(x,y,z,t) = 1$. For a deposition measurement, $K$ is
two dimensional with units $[L]^{-2}$ and the condition $\int \mathrm{d}t \int_{\mathcal{D}_2} \mathrm{d}x\mathrm{d}y K(x,y,t) = 1$, where $\mathcal{D}_2$ is the surface of the domain $D$.
In Eq. (12) the adjoint source function $\tilde{S}$ for an air concentration measurement simply corresponds to the kernel function $K$.
We can account for linear functions of the concentration field, such as deposition, by absorbing the proportionalities into the
adjoint source function. Thus, for wet and dry deposition the adjoint source function also contains the scavenging coefficient
$\Lambda$ (Seinfeld and Pandis, 2006) and dry deposition velocity $v_{\mathrm{dry}}$, respectively. Dry deposition is applied over a vertical distance
$h$, which is set to 30 m in Flexpart. The sampling time $T$ is also present in the adjoint source functions of both deposition
quantities in order to convert deposition rate to total deposition. Given these definitions, the adjoint source functions $\tilde{S}_i$ can be
interpreted as the effective geotemporal sampling space of each measurement, as schematically shown in Fig. 2. The adjoint
source function of a dry deposition measurement is similar to one of air concentration, as it samples air in the lowest part of
the boundary layer where material is deposited at a rate proportional to the deposition velocity $v_{\mathrm{dry}}(x,y,z,t)$. Wet deposition
provides a different vertical resolution compared to air concentration or dry deposition. Under wet deposition, the concentra-
tion is scavenged at the rate of the local scavenging coefficient $\Lambda(x,y,z,t)$, over the entire vertical up to the height of the
precipitating cloud. Thus, wet deposition provides a more extended vertical sampling space compared to dry deposition and
air concentration. This vertical extent can be beneficial in certain scenarios. Material will still be deposited if the plume passes
at an altitude but not near the surface, where it would miss an air concentration or dry deposition detector. Besides the vertical
resolution, the timing of the sampled air also differs significantly. Air concentration and dry deposition are sampled during
the entire measurement window, while wet deposition is sampled in precipitating conditions only, which can cover merely a
part of the full measurement window. Thus, a wet deposition measurement can potentially provide better temporal resolution
compared to an equivalent air concentration or dry deposition measurement. The benefits of this improved temporal resolution
with respect to inverse modelling will depend on the accuracy of the precipitation in the meteorological data used in the ATM
calculation. The net effect of the differences in vertical and temporal resolution is not considered trivial.

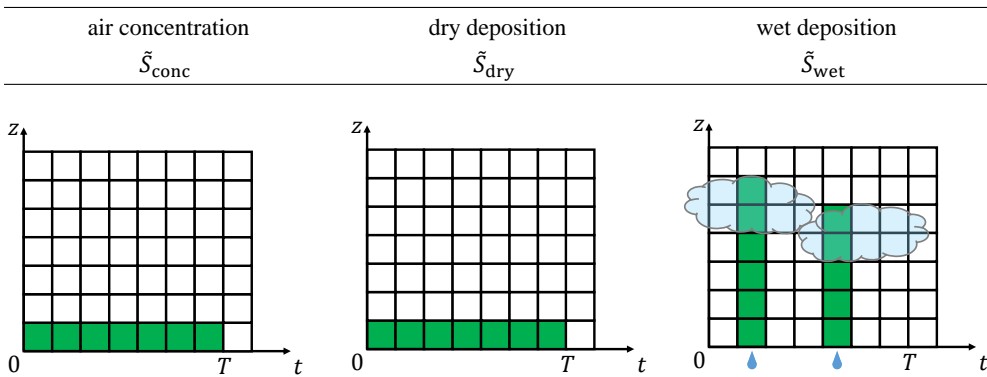

**Figure 2.** Schematic representation of the adjoint source functions $\tilde{S}_i(x,y,z,t)$ (green filled boxes) of air concentration, dry deposition and wet deposition measurements. $\tilde{S}$ represents both the geotemporal coordinates of the air sampled in each measurement, as well as the coordinates of the particle releases in the backwards-in-time dispersion calculations (Seibert and Frank, 2004; Eckhardt et al., 2017). The sampling starts at $t = 0$ and ends at $t = T$. The raindrop symbols indicate precipitation events, where wet deposition is collected.

The above-mentioned physical differences between the various types of measurements also have further implications for backwards-in-time calculations with a Lagrangian particle ATM, since $\tilde{S}$ is also the source function of the adjoint transport equation. As implemented in Flexpart (Seibert and Frank, 2004; Eckhardt et al., 2017), particles are released according to the adjoint source function and then evolved with the adjoint model. This means that the backwards-in-time method applied to air concentration and dry deposition measurements is expected to be similar, as particles are released continuously over the measurement window, close to the surface. For a wet deposition measurement, on the other hand, particles are released across the vertical. With the implementation in Flexpart, this dilutes the number of particles compared to the near-surface releases of air concentration and dry deposition. This can increase stochastic uncertainty in the model output as fewer particles are available to construct the retro-plume. Furthermore, in a backwards-in-time simulation particles that are allocated and released in non-precipitating conditions are immediately removed from the simulation, thus further reducing the output statistics. As mentioned in Sect. 2.2, the number of particles allocated for each simulation in this study is 10 million, which was determined based on practical time-constraints.

## 2.5 Performance metrics

In evaluating the performances of the different inverse modelling experiments, we will make use of the three performance metrics introduced by De Meutter et al. (2024). These metrics have been proposed to quantify the performance of various source localisation methods. These are schematically shown in Fig. 3 (adapted from De Meutter et al. (2024) with a license agreement):

(a) Distance

(b) Fraction of the domain excluded (FDE) $\in [0, \mathbf{1}[$

(c) Cumulative distribution score (CDS) $\in [0, \mathbf{1}]$

The distance metric quantifies the great circle distance between the true source location and the most probable location assigned by the inverse modelling method. The FDE is the fraction of the domain that is excluded as a possible source location based on threshold values: 2 for the residual cost and 0 for the Bayesian source location probability. It has a value between 0 and 1, with the latter being the perfect value. The total domain we consider is that defined by the grey rectangle in Fig. 1. The CDS is the value of the cumulative distribution function at the true source location. It also has a value between 0 and 1, with the latter being the perfect value. It can be defined relative to the full domain, or relative to a sub-domain defined by the coverage of the location probability. In this study, we use the definition using the sub-domain.

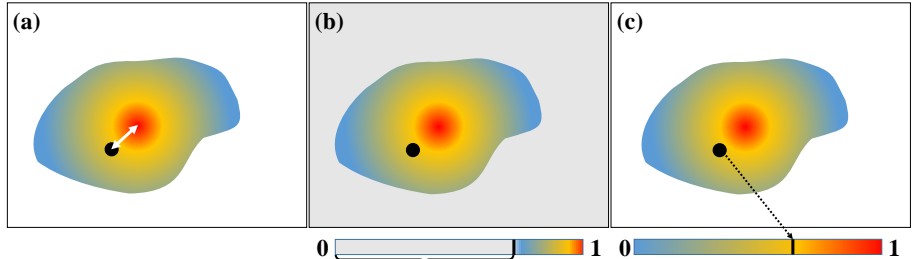

**Figure 3.** Schematic illustration of the three performance metrics as introduced by De Meutter et al. (2024) and adapted therefrom with a license agreement. (a) Distance (b) fraction of the domain excluded (FDE) and (c) cumulative distribution score (CDS). The black bullet represents the true source location and the coloured fields represent the source location probabilities.

## 2.6  Source term & synthetic observations

In this study, the inverse modelling techniques are evaluated with two different types of experiments: a 'twin experiment' and a 'real world' experiment. The twin experiment involves an inverse modelling calculation based on measurements generated from a forward ATM calculation. This type of experiment eliminates measurement, meteorological and model errors. The forward Flexpart calculation in our experiment is based on the [106]Ru source term from Saunier et al. (2019), as this is a source term from the literature with a described temporal release profile. They estimate a total release of about 250 TBq ($2.5 \cdot 10^{14}$ Bq), mainly released on 26 September (Fig. 4). This ATM calculation is then used to generate synthetic observations. The synthetic observations are subsequently used as the basis for the inverse modelling calculation, together with the SRS fields obtained from backward-in-time ATM calculations. As first glance, one might expect that the inversion results from such a type of experiment are somewhat trivial. This is, however, not necessarily the case since the synthetic measurements have a certain spatial and temporal resolution as explained in Sect. 2.4. Thus, a perfectly accurate result should not be expected, even for a twin experiment. It is also worth noting that the source term parameterisation used for the Bayesian inversion can only resolve a constant release and would thus not be capable of fully reproducing the profile as shown in Fig. 4. It will likely focus on fitting the main release on 26 September.

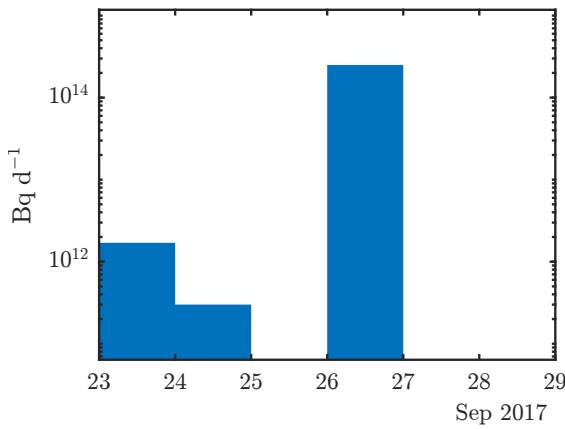

**Figure 4.** [106]Ru source term (Saunier et al., 2019) used for the forward ATM calculations in the twin experiments.

The synthetic observations are based on the real observations (Fig. 1) by using identical station locations and measurement windows. The synthetic data do not, however, come with minimal detectable quantities (MDQ). Nevertheless, these values are required for the inverse modelling algorithms (cf. Sect. 2.3). Thus, a choice has to be made. For the synthetic deposition datasets, an MDQ of 0.1 Bq m$^{-2}$ was chosen, reflecting (in order of magnitude) the lowest MDQ seen in the real dataset. Since air concentration (activity per unit volume) is a different quantity to deposition (activity per unit area), a different MDQ needs to be chosen as well. The MDQ for air concentration is also known as the minimal detectable concentration (MDC). The choice was made to use an MDC of 1 µBq m$^{-3}$, a value common for modern particulate monitoring stations.

## 2.7 Experiments

As mentioned before, we perform two experiments: the twin experiment and the real-world experiment. Each experiment contains multiple sub-experiments wherein different sets of measurements are used for the inverse modelling calculations. The experiment using real data contains three sub-experiments (Table 3). The rain water and fallout deposition datasets form the basis of two sub-experiments, with the third based on combining these two datasets. The twin experiments are set up differently (Table 4). Synthetic observations are generated for the 18 rainwater and 12 fallout measurements locations and windows. Since these are synthetic observations, the detected quantity can be chosen. Synthetic observations are generated for wet, dry and total deposition, and air concentration. In this way, a comparison can be made between the differences in vertical and temporal resolution of the deposition and air concentration observations (cf. Sect. 2.4). A fifth twin experiment is a combination of the total deposition and air concentration synthetic datasets with Eq. 8. The total deposition twin experiment uses the same measurements as the "rain water + fallout" real experiment and can thus be used in a direct comparison.

**Table 3.** Datasets used in the real data experiments. The circle and triangle symbols signify sets of measurement locations and windows, using the symbology from Fig. 1. The SRS fields follow the notation used in Eq. (8).

| real data | # | locations | SRS field(s) |
|---|---|---|---|
| rain water | 18 | ● | $M_{\text{tot}}$ |
| fallout | 12 | ▲ | $M_{\text{tot}}$ |
| rain water + fallout | 30 | ● + ▲ | $M_{\text{tot}}$ |

**Table 4.** Datasets used in the twin experiments. The circle and triangle symbols signify sets of measurement locations and windows, using the symbology from Fig. 1. The SRS fields follow the notation used in Eq. (8).

| twin experiment | # | locations | SRS field(s) |
|---|---|---|---|
| wet deposition | 30 | ● + ▲ | $M_{\text{wet}}$ |
| dry deposition | 30 | ● + ▲ | $M_{\text{dry}}$ |
| total deposition | 30 | ● + ▲ | $M_{\text{tot}}$ |
| air concentration | 30 | ● + ▲ | $M_{\text{conc}}$ |
| total dep. + air conc. | 60 | $2 \times (● + ▲)$ | $M_{\text{tot}}, M_{\text{conc}}$ |

## 3   Results & discussion

The results are organised in three sub-sections. Firstly, the model setup is evaluated by analysis of the deposition datasets with forward ATM calculations. Then, the inverse modelling results with the synthetic datasets are shown and discussed. Finally, the likewise is done for the inversion results with the real datasets.

### 3.1   Model setup evaluation

Figure 5 compares the the observed rain water and fallout deposition values with the modelled values obtained with the $(0.1°,$ 1 h) meteo data. All modelled rain water deposits fall within a factor 10 (FAC10) of the observations (Fig. 5, top panel and Table 5). Overall, the model consistently underestimates the observed deposition, as quantified by the fractional bias (FB) of $-0.64$. The correlation can be considered high with a Pearson coefficient $(R)$ of 0.85. The fallout measurements exhibit some different characteristics (Fig. 5, bottom panel). All model values fall within a factor 5 (FAC5) of the observations. However, the correlation is poorer, at 0.43. The reasons for the poorer correlation of the fallout dataset are unclear. It can be due to the collection method or other sources of error such as the source term or the model itself.

Almost all non-detections are reproduced perfectly, suggesting no erroneous rain event was captured due to increasing the measurement window as discussed in Sect. 2.1.

As described in Sect. 2.1, both the rain water and fallout datasets are assumed to contain dry and wet deposition. All simulated values for the rain water dataset contain a non-zero amount of wet deposition, while there are three fallout values that contain

no wet deposition. This difference supports studying the two datasets separately in our analysis. We will also discuss the results of combining both datasets.

There are two outliers in the modelled deposition values in rain water: labelled Sweden 3 and 6. They are underestimated by a factor 6.2 and 9.4 respectively. These data points have required some extra attention since it concerns two of the highest measurements.

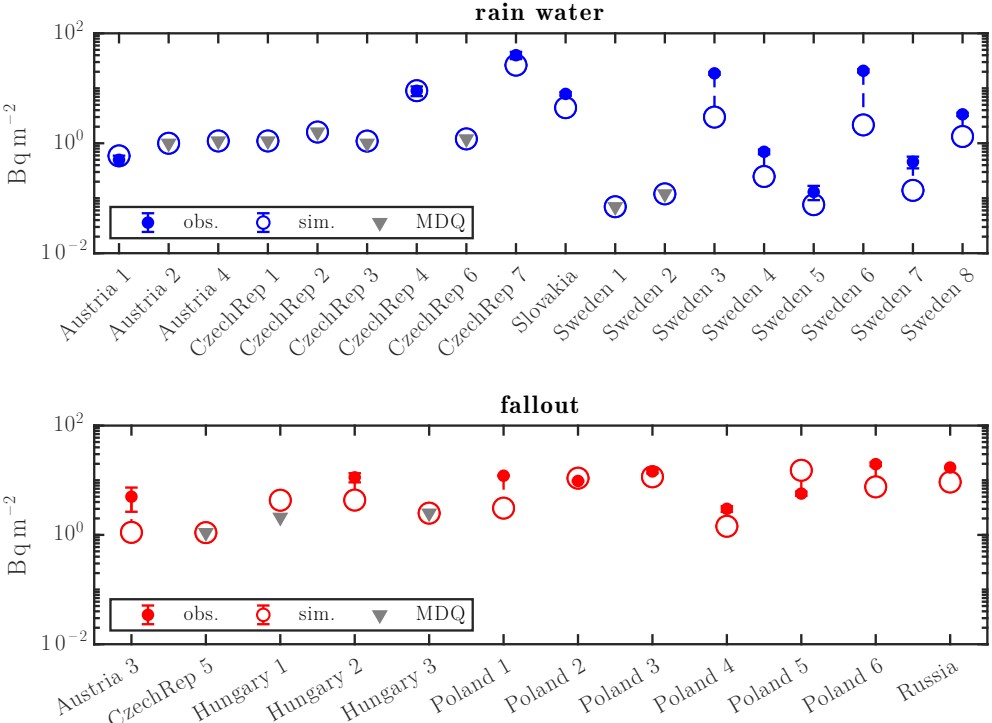

**Figure 5.** Comparison of observations and Flexpart model values based on the source term of Saunier et al. (2019) (top: rain water deposition measurements, bottom: fallout deposition measurements). Simulated values that fall below $\mathrm{MDQ}/2$ are artificially set equal to the MDQ for visual aid.

**Table 5.** Selection of statistical scores between the model and observed deposition values, for the two datasets: "rain water" and "fallout".

|  | FB | $R$ | FAC2 | FAC5 | FAC10 |
|---|---|---|---|---|---|
| rain water | $-0.64$ | 0.85 | 0.67 | 0.89 | 1 |
| fallout | $-0.33$ | 0.43 | 0.42 | 1 | 1 |

The observations Sweden 3 and Sweden 6 were made around 150 km from each other (in Gävle and Stockholm, Sweden) with measured values of $18.7 \pm 1.1$ and $20.8 \pm 1.5$ Bq m$^{-2}$ respectively. They are the second and third highest measurements

in the dataset. For both values, the model apportions around 35% to wet deposition and the other 65% to dry deposition. A comparison with the values obtained using the (1°, 3 h) meteorological data shows these measurements to be very sensitive to the resolution. This is shown in Figure 6. Using the lower resolution meteorological data, the two observations are underestimated by two orders of magnitude. The higher resolution meteorological data (0.1°, 1 h) provides an increase in deposition by one order of magnitude, which is still an underestimation but an improvement over the lower resolution result.

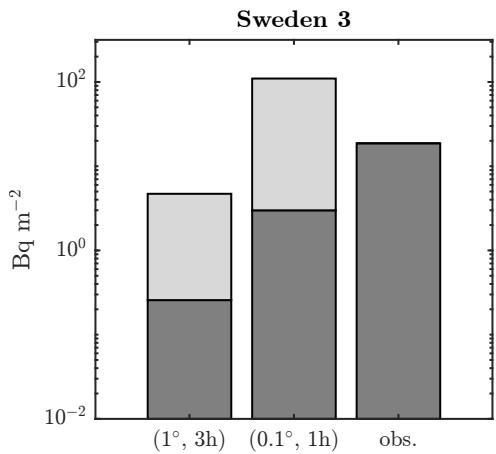 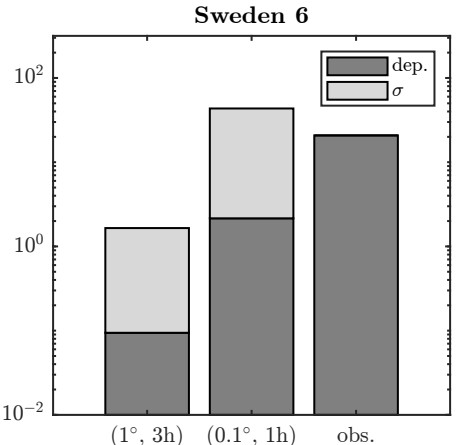

**Figure 6.** Modelled and observed deposition values for the Sweden 3 and 6 measurements. Modelled values use meteorological data with a spatiotemporal resolution of (1°, 3 h) and (0.1°, 1 h). The accumulated column density of a tracer species is denoted by $\sigma$.

In order to assess whether the remaining discrepancy is related to deposition or transport, we perform ATM calculations with an air tracer species. The air tracer species experiences no deposition, thus isolating the effects of transport. It is then interesting to analyse the column density $\sigma$, which is defined as the vertical integral of the air concentration field:

$$\sigma(x,y,t) = \int_0^\infty c(x,y,z,t)\mathrm{d}z, \tag{13}$$

and represents the total activity present in the vertical column. Taking the cumulative sum of the column density over time
gives a theoretical limit on the total deposition that could have occurred over that time. At any given point in time, all that can theoretically be deposited is that which is present in the vertical column. Accumulating this quantity over time then gives an upper limit on the accumulated deposition. Accumulating the column density in this way neglects any depletion that may occur in the plume from one timestep to the next and the fact that material will not be scavenged over the entire vertical, which should thus lead to an overestimation of the total possible deposition.
The comparison of the accumulated column densities between the (1°, 3 h) and (1°, 1 h) meteodata is also shown in Fig. 6 alongside the deposition values. As already described, the modelled deposition values with the (1°, 3 h) meteo fields are, for both measurements, two orders of magnitude too low. The column densities show that this cannot be explained by the calculation of deposition itself, as the column densities also fall below the observed values (despite being a very conservative

estimate). Thus no increase of deposition in this ATM calculation can possibly reproduce the observed values. The (0.1°, 1
340  h) calculation fares better. The deposition values have increased by around one order of magnitude compared to the (1°, 3 h)
calculation. This time the column densities actually exceed the observed deposition values. Figure 7 shows the spatial pattern
of the deposition accumulated over the measurement period. The location of the Sweden 3 and 6 measurements are denoted by
the black circles. For the (1°, 3 h) meteo data, the measurements are located along a relatively strong gradient, compared to the
(0.1°, 1 h) meteo fields where the gradient is much lower at those locations. This is the result of the air concentration plume
passing over Scandinavia being narrower in the (1°, 3 h) simulation, and containing a lower concentration overall.

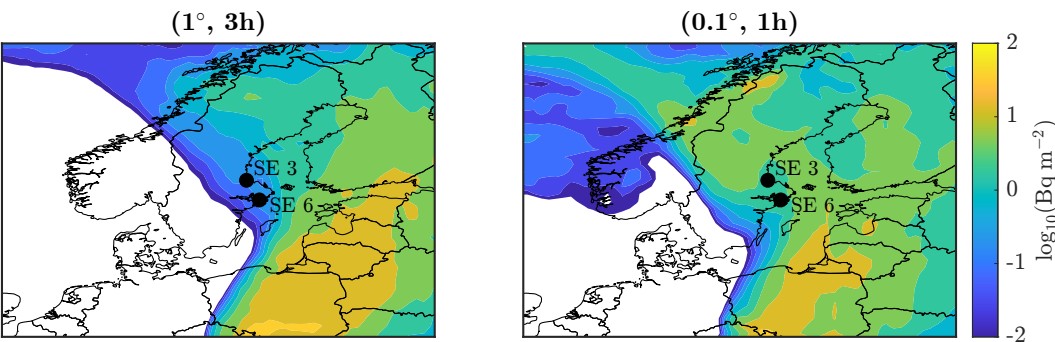

**Figure 7.** Total deposition using meteorological data with a spatiotemporal resolution of (1°, 3 h) and (0.1°, 1 h). The location of the Sweden 3 and 6 measurements are marked by the black circles.

No significant difference in deposition between the two sets of meteorological data was seen for other measurements. For this reason we continue with the (0.1°, 1 h) data.

### 3.2  Twin experiments

In Sect. 2.7 we have defined five datasets for the twin experiments. These experiments use synthetic data to eliminate measure-
ment, meteorological and model errors.

### 3.2.1  Deposition data

Figure 8 shows the source localisation results using the synthetic datasets of wet, dry and total deposition. The true source (Mayak, black circle) is located in a region of high probability for each of the three datasets. The performances can be further quantified by the three performance metrics introduced by De Meutter et al. (2024) (Sect. 2.5). These are shown graphically
in Fig. 9. The figure shows the three performance scores on three axes, though these should not necessarily be considered orthogonal (i.e. the metrics are not wholly independent). The choice was made to use $1 - \text{CDS}$ and $1 - \text{FDE}$ so that a value of zero represents the best score for all three metrics. The maximum limit of the distance axis is arbitrary, and chosen to be 1500 km, a distance that roughly corresponds to the largest possible distance to Mayak within the domain shown in Fig. 8.

The cost function method provides a very similar CDS and FDE for each twin experiment, with values around 1 and 0.7
respectively. The high CDS values signify that the true source location has a relatively low residual cost in each case. There
is a also an overall good fit between the synthetic and reconstructed values. The FDE's for the three experiments are similar,
despite the use of different deposition quantities in each dataset. The distance to the true source is around 500 km using the wet
and total deposition datasets. The dry deposition inversion appoints the most likely location to the correct grid-box. However,
this difference in distances just described is significant as may appear on first sight. The CDS's of the wet and total deposition
inversions are nearly equal to 1, hence, the location of the overall most probable location is only very slightly more probable
than that of the true source location. This emphasises some care needs to be taken in interpreting the performance scores.

The performance metrics for the inverse modelling with Bayesian inference are essentially perfect: a near zero distance
metric and a CDS and FDE near to 1 for each dataset. Nevertheless, there are some minor differences between the datasets
that can be identified on Fig. 8. The result with the wet deposition dataset shows, similarly to the cost function method, three
unconnected regions of local maximal probability. The Bayesian inference is able to assign a lower probability to the patches
west of Mayak compared to the cost function method. Using the dry deposition dataset, the previous most western patch is
excluded as a probable source region by the Bayesian inference. The dry deposition SRS components carry this property to the
total deposition results as well.

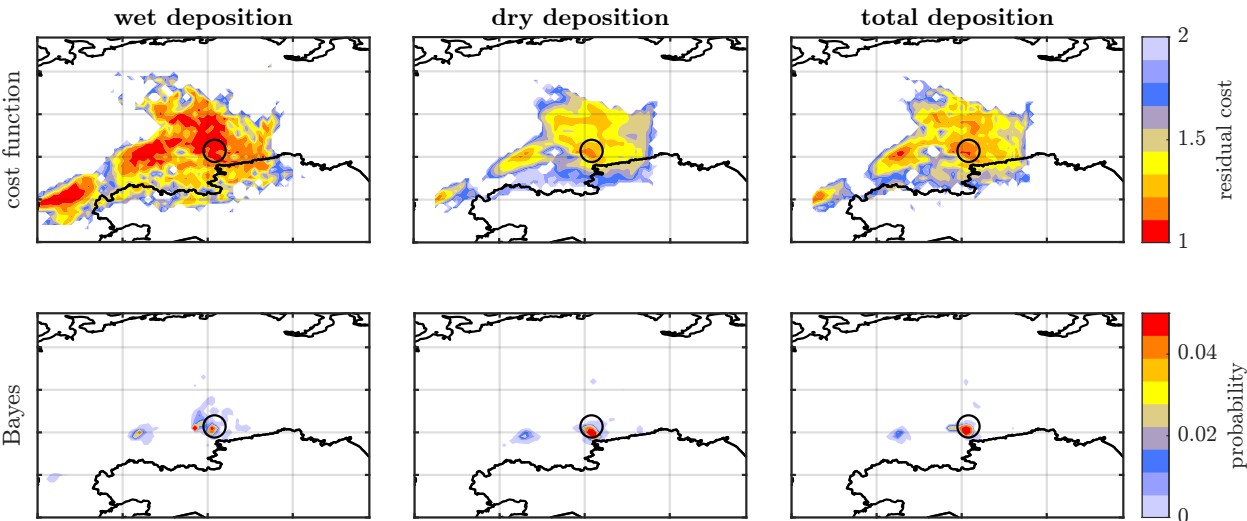

**Figure 8.** Top: residual cost after optimisation. Bottom: source location probability from Bayesian inference. Black circle: true source location
(Mayak). Left column: wet deposition twin experiment, centre column: total deposition twin experiment, right column: total deposition twin
experiment, as defined in Table 4.

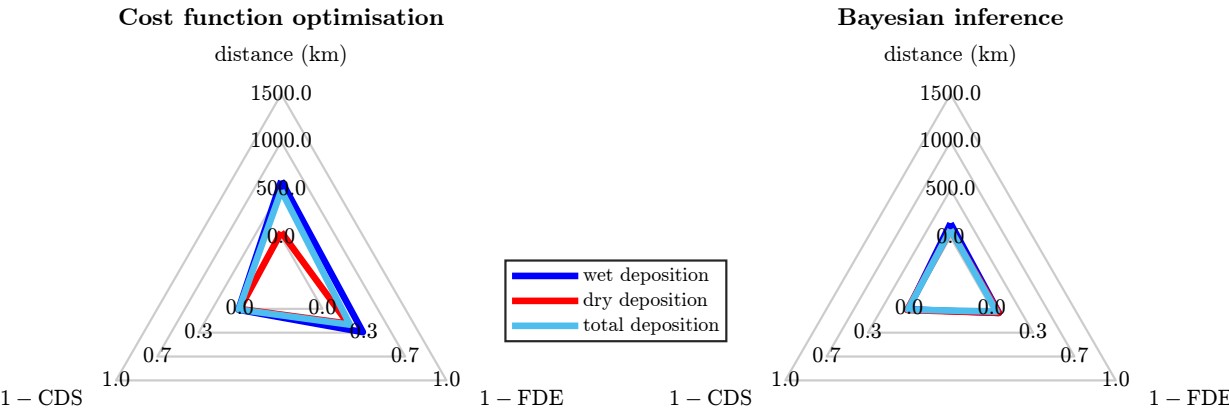

**Figure 9.** Performance scores of the cost function optimisation and Bayesian inference methods for the synthetic datasets: wet, dry and total deposition as defined in Table 4.

Besides the source location, the profile of the release (i.e. the released quantity over time) can also be of interest. This
information exists in different formats for each of the two inversion methods. The cost function method provides a release profile for each grid-box since the cost is minimised for each grid-box. The Bayesian inference method provides the (marginal) posterior distributions of each source term parameter, covering the whole domain. Therefore, an additional Bayesian inference inversion is performed for each experiment with the location fixed at the location of Mayak. In this way, the source terms obtained through cost function and Bayesian inference can be compared directly.

Figure 10 shows the release profiles of the three synthetic deposition datasets using the cost function method. These are the source terms as obtained within the grid-box of the true source location (Mayak). The inversion with dry deposition SRS fields is able to isolate the exact date of the major release. Using wet deposition SRS fields, a partial release is found on the correct day, and a day earlier. This effect is propagated when summing the wet and dry deposition SRS fields in the total deposition experiment. There, the release on the correct date is closer in magnitude to the true value. The algorithms are unable
to reconstruct the small releases on 23 and 24 September as they are several orders of magnitude smaller than the main release. These releases thus have a small effect on the deposition values, considering that the SRS values for these releases have the same order of magnitude (or lower) than the later releases. Figure 11 shows the release profiles of the three synthetic deposition datasets using the Bayesian inference method. The start time of the wet deposition experiment is around one day too early. The dry and total deposition experiments both show a similar start time signal, which is closer to the real value and show a
clear cut-off after 26 September. The end times of all three experiments show a signal around the correct date. The release magnitudes $Q$ of all experiments is overestimated by a factor of around 2. The fractional bias of the best fit is, however, close to zero ($< 0.1$) for all three experiments. The overestimation is then likely a cause of a combination of factors. The Bayesian algorithm may assign different start and/or end times to the release, where a larger source term is found. trying to compensate

for the small inconsistencies between forward and backwards Flexpart simulations. These inconsistencies can arise due the

small difference in interpolation of the meteorological input data (Seibert and Frank, 2004; Eckhardt et al., 2017).

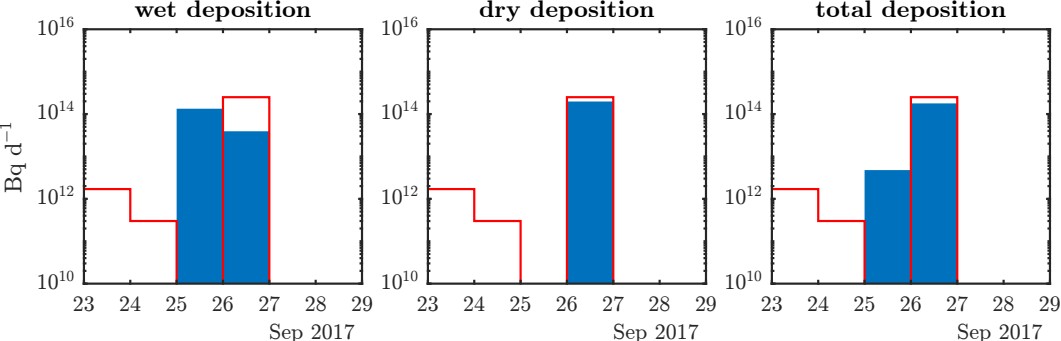

**Figure 10.** Optimised source terms following cost function optimisation in the Mayak grid-box, for the deposition based twin experiments as defined in Table 4. The red outline is the true source term (Saunier et al., 2019) used for generating the synthetic observations.

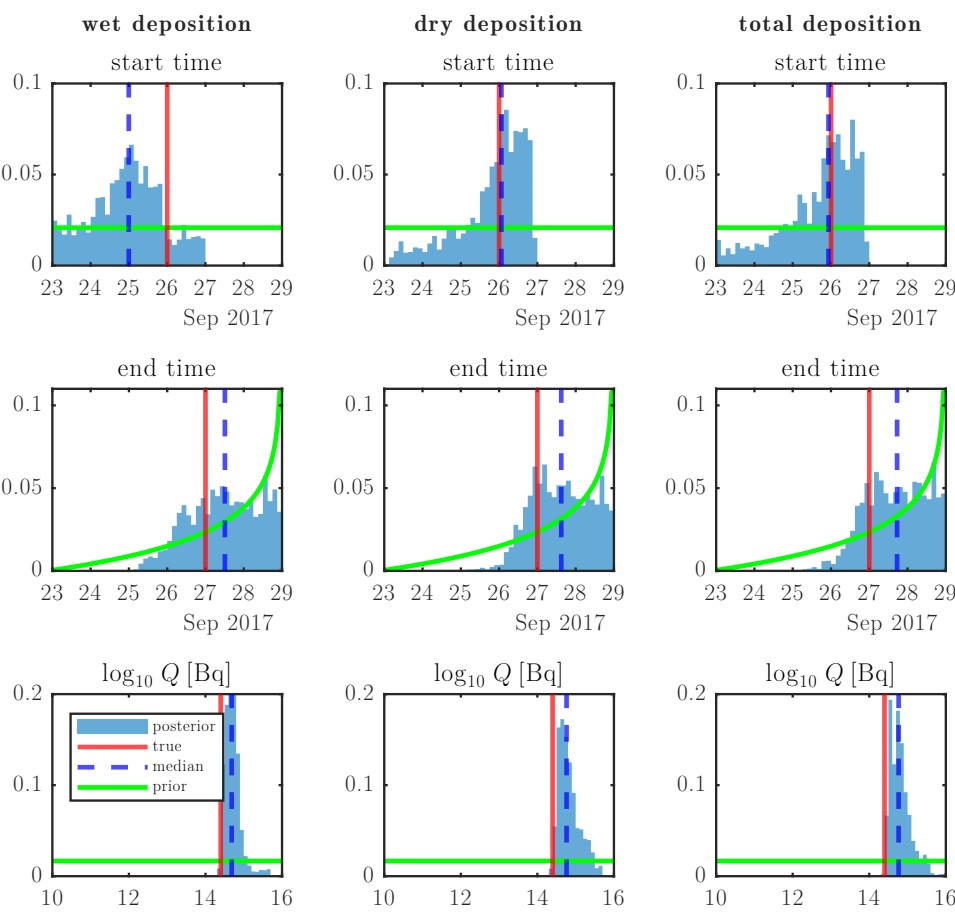

**Figure 11.** Probabilities of source term parameters following Bayesian inference, with each deposition twin experiment as defined in Table 4.

### 3.2.2 Air concentration data

As described in Sect. 2.7, we also constructed a twin experiment using a synthetic dataset of air concentration measurements. These synthetic measurements have the same location and measurement windows as the deposition data. Figure 12 shows the inverse modelling results of the air concentration twin experiment and the experiment combining all SRS fields using
Eq. (8) (total deposition + air concentration experiment in Table 4). The results for both cases are extremely accurate and precise. Figure 13 shows the corresponding scores. All inversion experiments that contain air concentration SRS fields can be considered quasi-perfect. Only the CDS using the Bayesian inference deviates from perfection with a value of around 0.7.

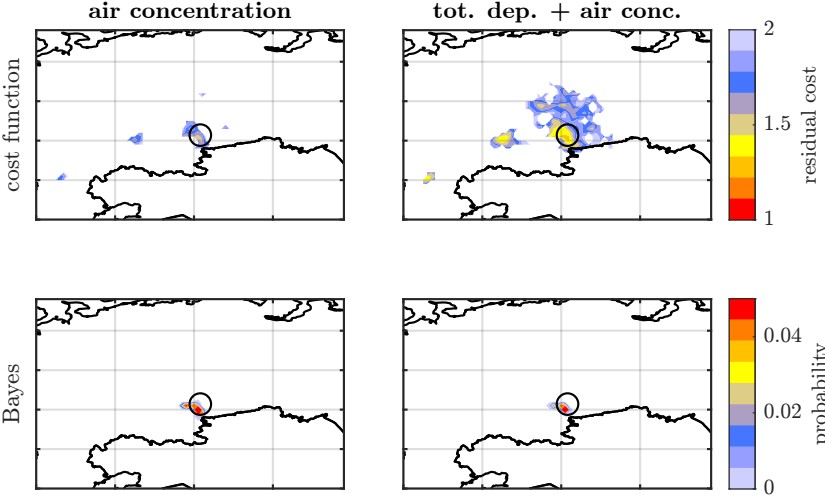

**Figure 12.** Top: residual cost after optimisation. Bottom: source location probability from Bayesian inference. Black circle: true source location (Mayak). Left column: air concentration twin experiment, right column: total deposition + air concentration twin experiment, as defined in Table 4.

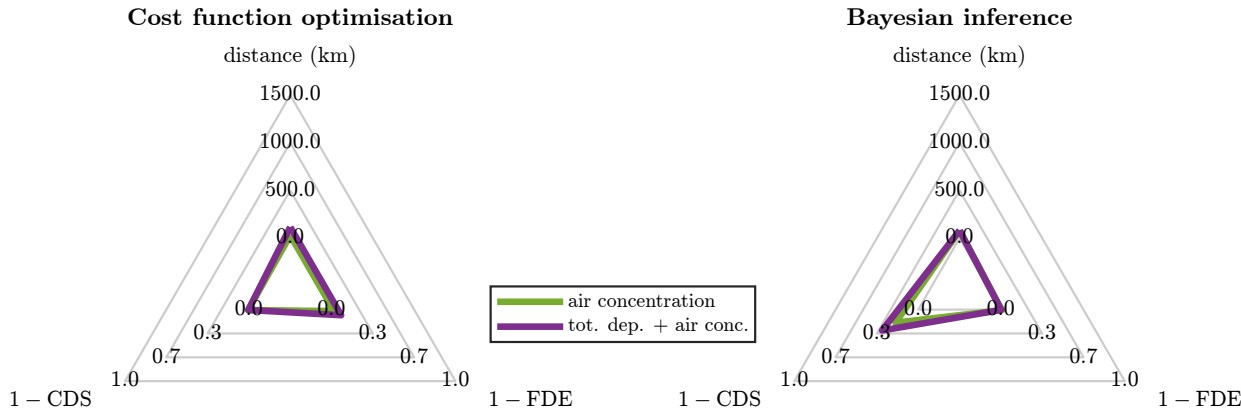

**Figure 13.** Performance scores of the cost function optimisation and Bayesian inference methods for the synthetic air concentration datasets as defined in Table 4.

The release profile using Bayesian inference in the air concentration experiment, shown in Fig. 14, provides an interesting comparison with the dry deposition experiment from Fig. 11. Since both dry deposition and air concentration SRS fields are very similar (see Fig. 2), the inverse modelling results are expected to be similar. This is verified with the results as shown.

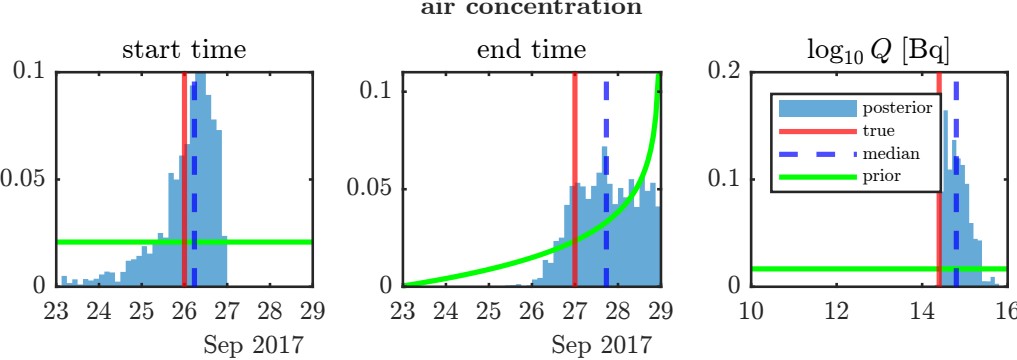

**Figure 14.** Probabilities of source term parameters following Bayesian inference with the synthetic air concentration dataset.

It can be considered peculiar that the source localisation using the cost function method is much better in the air concentration experiment compared to the deposition experiments. This is most clearly expressed by the FDE's: 0.7 for the deposition experiments (Fig. 9) versus 0.99 for the air concentration experiment (Fig. 13). Further analysis shows this can be mainly attributed to the difference in the MDQ's. The (synthetic) values of deposition are, in relative terms, closer to the chosen

MDQ (0.1 Bq m$^{-2}$) than the air concentrations are to their MDC (1 μBq m$^{-3}$. The air concentration values are generally much larger than this MDC. Re-running the air concentration experiment with an increased MDC of 0.1 mBq m$^{-3}$ yields the results shown in Fig. 15. The FDE is now comparable to those of the deposition experiments. The overall shape of the residual cost is particularly similar to that of the dry deposition experiment (Fig. 15, middle panel, top row). This is expected, as dry deposition and air concentration measurements sample similar parts of the plume (cf. Sect. 2.4). The MDC of 0.1 mBq m$^{-3}$, however, can

be considered too high for modern technologies. We thus find that, theoretically, the largest influence on source localisation is not the type of measurement, but rather the detection limits thereof.

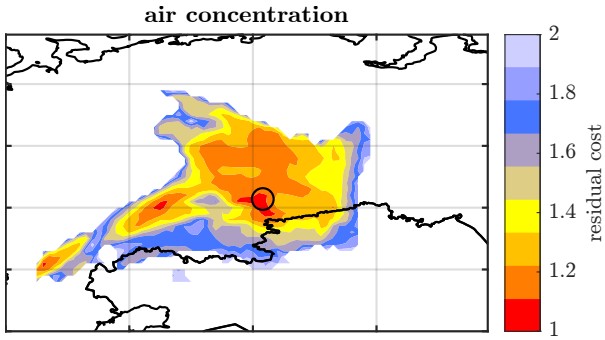

**Figure 15.** Residual cost after optimisation using synthetic air concentration measurements with an MDC of 0.1 mBq m$^{-3}$. Black circle: true source location (Mayak).

## 3.3 Real data

In this section, the inverse modelling techniques are applied to the real data. When evaluating the performance scores in this section, it is assumed that the Mayak nuclear installation is the true source location.

The source localisation results of the cost function and Bayesian inference methods are shown in Fig. 16 and their scores in Fig. 17. Significant differences can be seen between the rain water and fallout inversion experiments. The cost function method shows a better localisation with the rain water data compared to the fallout data. The fallout data covers a larger part of the domain, as quantified by the FDE of the fallout experiment being 0.5 compared to 0.7 for the rain water data. The residual cost at Mayak with the fallout data is greater compared to the rain water data, implying the SRS fields are better able to reproduce the latter measurements. The distance metric for the fallout data is also rather large, at 1500 km. This is the most western patch of local minimal cost visible on the figure. However, the local minimal cost neighbouring Mayak is only very slightly higher, as reflected in the CDS of $\sim 1$. Thus, this should not necessarily be considered a bad result. The Bayesian method is able to assign a lower probability to this westernmost patch, placing the most likely source location at around 300 km from Mayak, in the close-by region of high probability. The Bayesian inference with rain water data is overconfident compared to the cost function method, a property of the Bayesian method in FREAR which has been observed in previous studies (De Meutter et al., 2024). The combination of the rain water and fallout datasets is shown in the column "all data". This can be directly compared to the total deposition twin experiment of Fig. 8 (right panels), as they use the same SRS fields. It is somewhat remarkable then, that both results appear very similar. The Bayesian inferred CDS of the real data experiment is however much smaller compared to its synthetic counterpart. This is due to the over-confidence of the Bayesian method. Nonetheless, in a real-world case with truly unknown source, one would still be able to identify Mayak as the only close-by nuclear installation, without ambiguity.

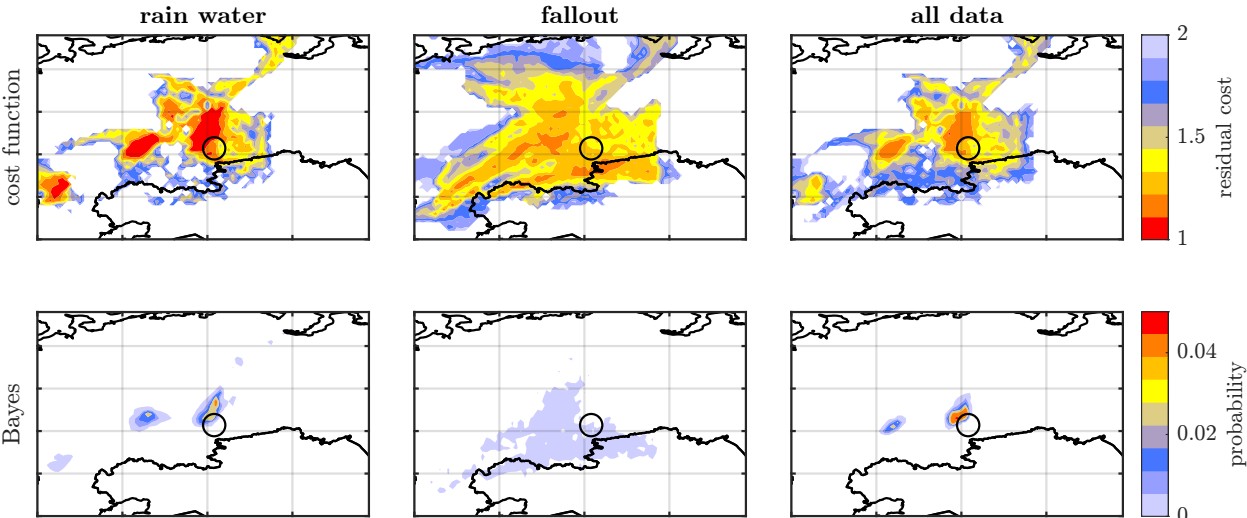

**Figure 16.** Top: residual cost after optimisation using real deposition measurements. Bottom: source location probability from Bayesian inference. Black circle: location of the Mayak nuclear installation. Left column: using rain water measurements, centre column: fallout measurements, right column: rain water + fallout deposition measurements, as defined in Table 3.

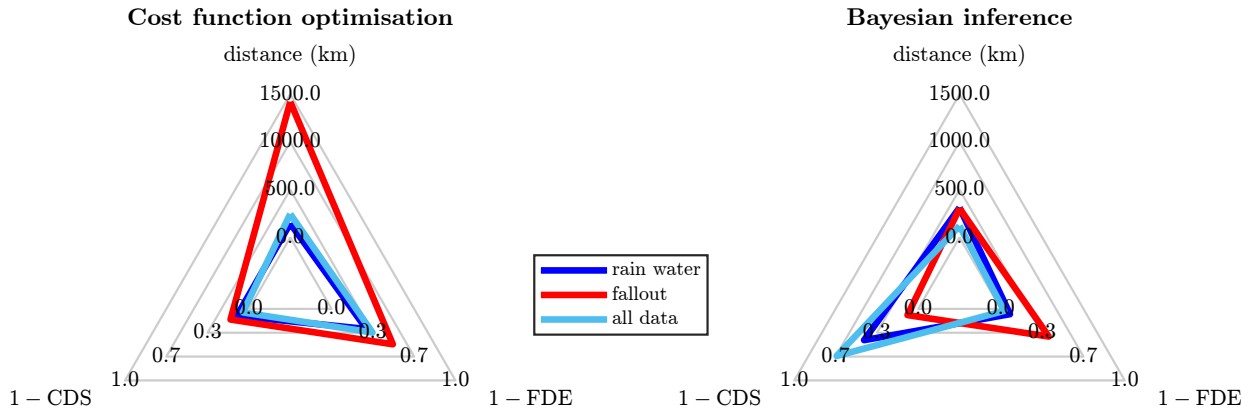

**Figure 17.** Performance scores of the cost function optimisation and Bayesian inference methods for the real deposition datasets: rain water, fallout and rain water + fallout (all data), as defined in Table 3.

The optimised release profiles following cost function optimisation are shown in Fig. 18. The total amounts released are 350, 250 and 290 TBq for the rain water, fallout and the combination of both datasets respectively. All these values are comparable to the source terms from existing literature (Table 1). Using the rain water data, the release occurs fully on 25 September, while the fallout data also give a split release one day later, on 25 September. The timing of both these source terms falls within the

range of 24–26 September that is covered by the existing literature. Using both datasets simultaneously results in the algorithm assigning the release fully to 25 September. The release profiles following Bayesian inference are shown in Fig. 19. The rain water dataset leads to relatively well defined start and end times that are within 24h of that of Saunier et al. (2019). The release magnitude is about a factor 5 higher. The timing when using the fallout dataset fares less well. Both start and end times show

only weak signals, though a clear cut-off is seen in the start time, excluding a release start after 26 September. Combining the rain water and fallout datasets provides results that are close to that of the rain water dataset. This can be compared once more to that of its synthetic counterpart (total deposition of Fig. 11), showing remarkably similar results.

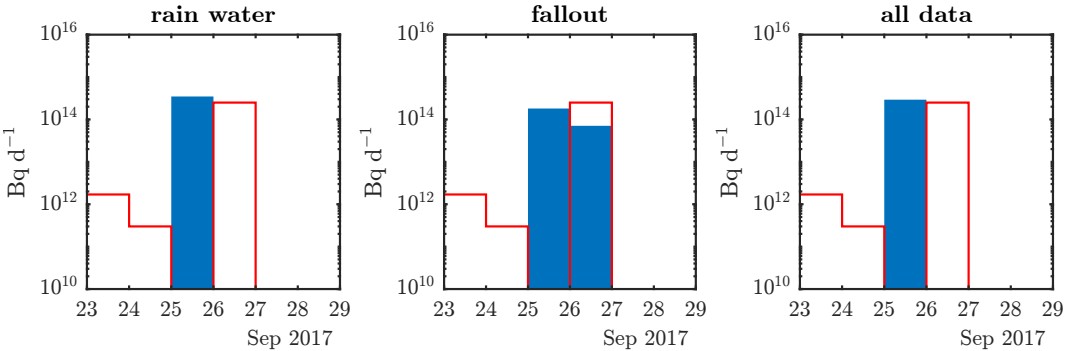

**Figure 18.** Optimised source terms following cost function optimisation in the Mayak grid-box, for each dataset as defined in Table 3. The red outline is the source term of Saunier et al. (2019).

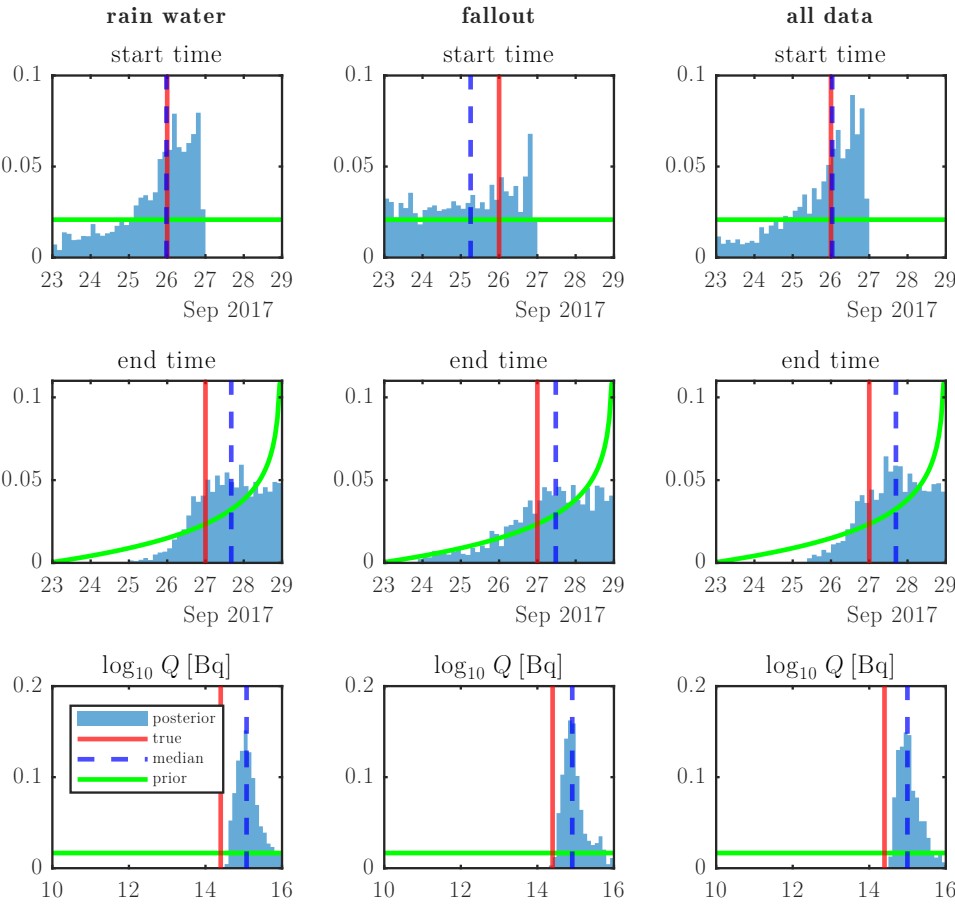

**Figure 19.** Probabilities of source term parameters following Bayesian inference with each dataset as defined in Table 3. 'True' values are based on the Saunier et al. (2019) source term.

While we have not used the >1000 air concentration measurements that are available for the [106]Ru case, we can speculate on the impact of combining them together with the deposition measurements. Since the total number of deposition measurements is one order of magnitude smaller – at around 100 – the resulting impact of combining the air concentration and deposition measurements is expected to be minimal since we have shown that a deposition measurement generally contains a similar amount of information as an air concentration measurement for the purposes of source reconstruction. There are also other studies which have combined air concentration and deposition measurements, albeit by assuming a known source location. Winiarek et al. (2014) and Dumont Le Brazidec et al. (2023) estimate the [137]Cs source term of the Fukushima nuclear disaster based on combining air concentration and deposition measurements. Dumont Le Brazidec et al. (2023) find that adding

deposition measurements leads to a significant improvement in the fit to the deposition observations, while their fit to the air concentration measurements remains similar. Winiarek et al. (2014) find that their algorithm has too much freedom to fit the data when solely using deposition measurements. However, the Fukushima release term is much more complex than the short [106]Ru release considered in this paper, with complex variations in strength over several weeks. We are able to obtain better

results using deposition measurements, presumably due to the (assumed) simpler source term.

Finally, we remark on the differences in performance between rain water and fallout deposition measurements in our study. Ultimately, the origins of these differences are unclear. While the fallout dataset contains less measurements, effects are seen that cannot be explained by this difference. The lower residual cost and probability mean that the reconstructed depositions also provide a poorer fit to the measurements. One possibility is that the fallout deposition measurements are somewhat poor,

or have an under-reported uncertainty compared to the rain water measurements.

## 4   Conclusions

We have investigated the use of deposition measurements for inverse modelling by applying it to the case of an undisclosed large release of [106]Ru in Eurasia during the autumn of 2017. The inversion was performed with two algorithms provided by the inverse modelling software FREAR: a cost function optimisation and a Bayesian inference method. Two types of inversion

experiments were set up: one using the real dataset of deposition measurements made in Europe, and one using synthetic observations. The real datasets consist of activity measured in rain water and fallout. The synthetic deposition datasets are comprised of synthetic wet, dry and total deposition measurements at the locations and with the observation windows of the real data. On top of that, we added a corresponding synthetic air concentration dataset with the same location and measurement timings as the deposition data. We found a large impact of the resolution of the meteorological data on two measurements in

Sweden, suggesting that high resolution meteorological data can help to improve the accuracy of source reconstruction.

The synthetic datasets provide a probe into the fundamental abilities of deposition measurements in inverse modelling by eliminating measurement and model errors. Inverse modelling using synthetic wet deposition measurements yields similar results to using the synthetic dry deposition measurements. This despite the fundamental difference in temporal and vertical resolution of these quantities. Comparing the synthetic deposition and air concentration datasets shows that one can expect

more precise results using air concentration data due to the relatively lower detection limits as the source localisation results exclude a larger fraction of the domain. From this, we conclude that lowering the detection limits of deposition measurement could aid source localisation with these measurements. Nonetheless, deposition measurements are generally cheaper and more versatile in practice compared to air concentration measurements. Deposition collectors can be placed in locations likely to be hit by an airborne plume, or ground samples can be taken after the passage of the plume.

The datasets with the real measurements provide results comparable to those of the synthetic datasets. The reconstructed release timings and magnitudes fall within the range found in existing literature. Using rain water measurements, the source localisation approaches that of the twin experiments. The fallout measurements, however, provide a somewhat worse results. This could be due to the fewer number of measurements for the fallout dataset, a difference in collection method between

both datasets, or model errors. Combining the rain water and fallout datasets in the inversion algorithms provides results closer
to those of the rain water dataset. From these results, we conclude that source localisation and reconstruction with deposition
measurements, be it wet, dry or total deposition, is feasible and can yield useful results in the context of radiological emergency
preparedness and response, and Nuclear-Test-Ban Treaty relevant events.

*Code availability.* The FLEXPART v10.4 source code is available at https://doi.org/10.5281/zenodo.3542278 (Pisso et al., 2019). The base
FREAR code is available at https://gitlab.com/trDMt2er/FREAR. The adjusted R scripts are provided at https://doi.org/10.5281/zenodo.
14525978.

*Data availability.* The deposition observations are contained in the Appendix of Masson et al. (2019). The datasets of all inversion experiments are available at https://doi.org/10.5281/zenodo.15594121 in `.mat` format.

*Author contributions.* SVL: conceptualisation, methodology, formal analysis, software, writing – original draft preparation, visualisation.
PDM: conceptualisation, methodology, writing – review and editing, supervision. JC: conceptualisation, methodology, writing – review and
editing, supervision. PT: conceptualisation, writing – review and editing, supervision. AD: conceptualisation, methodology, software and
meteodata access, computing resources, supervision

*Competing interests.* The authors declare that they have no competing interests.

*Acknowledgements.* The authors would like to thank the two anonymous referees and referee Isaac Ravi Jadav for reviewing our manuscript
and for providing fruitful feedback.

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
