# Peer review of "Source reconstruction via deposition measurements of an undeclared radiological atmospheric release"

_EGUsphere, 2024_

## Referee Comment (RC1)

**General comments**

The article presents clearly an important methodology concerned to radiological emergency preparedness. This methodology adopts cheaper resources than the traditional air concentration measurements. The work should be published.

**Specific comments:**

Following are the specific comments addressed to the work:

1) The Shershakov et al. (2019) reference on Table 1 is uncompleted. Please, complete or remove this reference from the Table.

2) Line 210: Table 1 shows highly different source terms. Why was the source term from Saunier et al. (2019) chosen? Were the others also tested?

3) Table 5 wasn´t called anytime in the text. Please, identify each acronym and explore better those scores. What does explain the better correlation for the rain water experiment? Lines 244-245.

4) Line 311: "***These releases thus have a small effect on the deposition values***". Is the released quantity the only effect? Could the atmospheric conditions like wind speed and direction or atmospheric stability simulated by the model impact this result? Could the first partial release detected by the inversion method related with the wet deposition (Sep, 25$^{th}$), in truth, refer to the release of the 2 previous days? Could the same release result on significant deposition values for a less favorable dispersion condition?

5) Line 392: "***We found an unexpectedly large impact of the resolution of the meteorological data in Sweden".*** Why unexpectedly? Since higher resolution modelling can simulate better the physical processes, especially within the Planetary Boundary Layer, the improvement is totally expected. Please, rewrite this phrase.

---

## Referee Comment (RC2)

**General comments**

I think the manuscript is overall well structured and describes a well conducted and thorough study. I believe that the article examines an important subject and nicely fills a gap in existing literature by systematically examining the use deposition measurements for source term estimation. Further, I think the approach of applying the method to both synthetic data and a real world case is a good way of demonstrating the capabilities and short-comings of the method.

Personally, I would have liked to see an experiment, where all the existing air concentration measurements were used, both with and without the deposition data (both for the synthetic and real data). I think it would be interesting to see if there is added value of combining the two datasets, or if deposition data is less important, when you have +1000 air concentration measurements already. I understand that adding this experiment might be outside the scope if this study, but perhaps a small discussion could be included, where the authors speculate a bit about this.

In addition, I believe that some points are a bit unclear and need further clarification. I have listed some specific comments below, which I think needs to be addressed before the manuscript is ready for publication. That said, I strongly support the publication of this manuscript, if these points are clarified.

**Specific comments**

1. **Line 95:** You write that only 30 are left for your analysis. I understand that 35 were discarded, because they were too close to the source, but earlier (line 95), you mentioned that there are 135 measurements in total. What happened to the remaining 70? You write that "… *only detections of activity per surface area (i.e. Bq m -2 ) were selected.*" What were the remaining, and why are they not useful?

2. **Line 95:** "*We follow the distinction made by Masson et al. (2019) to label 18 of these "activity concentration in rain water" and 12 "dry + wet fallout" … Though the description "rain water" may seem to imply only the collection of wet deposition, in general monitoring networks do not discriminate between dry and wet deposition. It is therefore assumed that both the rain water and fallout measurements contain dry and wet deposition …*"
   I am not sure I understand your approach here. You argue that you cannot justify making a distinction between the two datasets, and you therefore assume that both consist of dry+wet deposition. If this is your assumption, then why do you use the labeling? I find this is a bit confusing.
   I would probably have preferred one of two options **(a) make a distinction between the datasets, and then assume that "rain water" measurements consist of only wet deposition (then use that knowledge for the inversion)**, or **(b) don't make a distinction between the datasets and treat them equally (as you do), but then don't use the labeling and use only the combined dataset and get more robust statistics.**
   If you choose to keep your approach as it is, I think you should further justify why it makes sense to study the two datasets individually. Also, you should make it more clear that "rain water" is only a labeling and not related to the assumed deposition process. In the rest of the article, it is a bit unclear to me how you interpret this label yourself.

3. **Line 100:** "*The deposition data in the supplementary material of Masson et al. (2019) only provide the start and end dates of the measurements.*" I miss some information about typical durations of the measurement windows. I think this is a crucial information both for understanding the usefulness of the measurements in the first place, but also for understanding how large the potential error is on the assumed start and end times of the measurements.

4. **Line 105:** "… *the choice to extend the measurement interval by 8 h was made to increase the likelihood of capturing the relevant precipitation event that contributed to any wet deposition*." But that goes both ways. You may end up including a precipitation event that was in fact outside of the measurement window.

5. **Line 150:** Can you perhaps elaborate a bit on the Bayesian approach. When you write that you use "… *inverse gamma distribution for the combined model and observation uncertainties*", do you then mean as prior distribution for the uncertainties, which are assumed unknown? Please, specify this. Further, it would be nice to get a few details about the MCMC sampling method used.

6. **Line 220:** "*A relative measurement error of 50% was chosen.*" What do you use this for? Is it input to the inversion algorithms? If this is the case, do you assume that this represent only measurement error or combined measurement+modelling error? Finally, if the latter is the case, should you not use a different number for the synthetic vs. real case?

7. **Figure 5:** Here you show how the modelled dry+wet deposition correspond to the observations of two datasets "rain water" and "fallout". Again, I am not sure I understand the distinction, since you treat the measurements identically. You could increase the amount of data by combining the datasets, and obtain more robust estimates of statistical parameters such as correlation coefficient and fractional bias.
One thing that could be interesting is to see how the modelled dry and wet deposition alone correspond to the observations. Can you for example see that the dataset labeled "rain water" seem to be dominated by the modelled wet deposition.
If you do not include figures showing this, I would at least appreciate a section somewhere in the text, where you comment on the magnitude of the modelled wet vs. dry deposition values. If you can see that the wet deposition is dominating for the "rain water" dataset, I think this adds to the justification of studying the two datasets individually (cf. my comment nr. 2).

8. **Line 252:** "*The higher resolution meteorological data (0.1◦, 1h) provides an increase in deposition by one order of magnitude, which is still an underestimation but an improvement over the lower resolution result.*"
I think this is an interesting result, and I especially appreciate the following analysis, where you compare the accumulated column densities for the two different resolutions (Figure 6). However, I notice that both of these measurements are in the dataset labeled "rain water". For gaining further understanding, I think it would be relevant to know if the modelled deposition for these locations is mainly wet, as the labeling suggests (cf. my comment nr 7).
Further, did you look at differences in plume structure? Maybe there are some significant differences in the resolved flow? Or perhaps the measurements are taken in an area with large concentration gradients; then even small differences in the in the plume position could explain large discrepancies.

9. **Figure 7:** You have not really introduced the term "residual cost" before this figure. Can you perhaps elaborate a bit on the interpretation of this. Do you interpret it as being proportional to a probability density?

10. **Figure 9 and 10:** It is not until I see these figures that I can guess what type of assumptions you have made about the source term in the two inversion methods. For the cost function based method, you have a release profile with different release rates for each day, while for the Bayesian inversion, it seems that you assume a constant release over a period (described by start time, end time and release rate)? These assumptions should be stated clearly somewhere

in the text. Since you have decided to use the "twin experiment" with two separate releases as the "truth", it is especially relevant to mention that it is not possible to describe this with the source term discretization you chose for the Bayesian inversion.

Further, in Figure 10, I can see the prior distributions you have used for start time, end time and release rate, but I would like to read an explanation somewhere.

11. **Line 328:** "*Since both dry deposition and air concentration SRS fields are very similar (see Fig. 2), the inverse modelling results are expected to be similar. This is verified with the results as shown.*" I think this conclusion is very interesting, especially combined with the re-run as described in line 335.

12. **Line 339 and 400:** You conclude that lowering the detection limits of deposition measurement could aid source localization with these measurements. First of all, this statement is probably fair because lowering the detection limit can only improve the results.

However, I think you should be careful with concluding too much based on the idealized case with negligible model uncertainties. In "real world" applications, the model uncertainties are expected to be much larger than the measurement uncertainties, so I would not expect to see as big an impact.

It would be interesting to see it demonstrated on real data. One option could be to conduct the inversion with all the existing air measurement data (using the real data) and then artificially raise the detection limit of those to see what the impact would be. This is probably a task for a different study, but I think you should at least discuss what impact you would expect when using real measurement data.

13. **Line 358-360:** You describe this problem of the method being over-confident. And while I agree that you would easily be able to point out Mayak in the specific case, it is of course somewhat of a problem. Before I would consider using this method I would appreciate a discussion regarding the cause of the issue as well as possible solutions.

Could the problem be that the uncertainties are assumed too small? I am still not 100% sure how the uncertainties are treated in this Bayesian method, because you first mentioned the inverse gamma distribution, and then later wrote that you assume a 50% relative error on the measurements. (cf. my comments 5 and 6). So I look forward to elaborations on this.

14. **Figure 15:** The Bayesian method gives a very impressive result.

15. **Line 405:** "*The fallout measurements, however, provide a somewhat worse results for reasons that are unclear.*" To really discuss this, I still need clarification about why the dataset is split into these two subsets in the first place. If we knew that one dataset is dominated by wet deposition and the other by dry deposition, then I guess that would be the interesting part to discuss. However, if we assume that the two datasets are comparable, then one explanation could be that you have too little data. After all, the "fallout" dataset only consists of only 12 measurements. I hope that your answers to some of my previous questions can also help a bit with the understanding here.

---

## Referee Comment (RC3)

**General comments**

This paper studies a novel use of deposition measurements: the locating of an undeclared source. Source inversion in this context is usually a data sparse problem; the method presented enables more of the available data to be used a consistent manner, increasing confidence in inversion results. The method is systematically investigated and its presentation in this paper is clear and methodical.

Below, I have a few comments / questions which I think require consideration before the paper is ready for publication. If these are addressed, I encourage the publication of this work.

My thoughts largely overlap with the comments from the first two anonymous referees. Hence, I restrict my comments to those which differ from / extend theirs.

**Specific comments**

1. **Line 21.** *"Specifically, the release of radionuclides – or more precisely: its accompanying ionizing radiation – can potentially pose ..."*

   I think this sentence reads more clearly without the hyphenated expression ("- or more precisely ... ionizing radiation –") because the same sentence later states that the danger is from exposure to radiation.

2. **Line 119.** *"The retro-plume dispersion was calculated from the start of each measurement".*

   What does this mean? I understand that particles are released at a constant rate over the whole measurement window. If each particle starts being advected as soon as it is released, should this be "from the *end* of each measurement"?

3. **§2.4**. Comments / questions on the adjoint SRS calculations:
   a. Fig. 2 is nice, providing an intuitive explanation of the nature of wet / dry deposition observations, and their information content for the purposes of adjoint modelling. It complements the description of the method given in (Eckhardt et al., 2017) (if I am correct that it is the same method?). Since the Eckhardt paper contains many important technical details (e.g. height distribution and masses of model particles), please can you re-reference it in §2.4.
   b. For completeness, it would be nice for Fig. 2 to include the (adjoint) source functions for the SRS fields, $h(x, y, z, t)$, corresponding to each measurement type (these are proportional to $h$ defined by: $y = \int_{-\infty}^{\infty} (h \cdot c) \, d^3x \, dt$, where $c$ is the activity air concentration field, as in §3 of (Yee et al., 2008)). This essentially rewrites the integrals in a common inner-product framework. Then, I think Fig. 2 would be of even greater use for understanding the SRS method for deposition measurements; in particular, how it relates to the method for air concentration measurements.

4.  **§2.5**. Comments / questions on the performance metrics:
    a.  How is the 'excluded' part of the domain defined in the FDE metric? Is there a threshold for the cost function / Bayesian posterior value? Does the numerical solver / sampler simply fail in these cases?
    b.  The domain enclosed by the grey rectangle in Fig 1 should be described (e.g. using lat-lon coordinates. Perhaps in the caption to Fig 1?). Then, if others repeat this work but with different computational domains, they will be able to calculate an FDE score which is comparable with the present work.
    c.  *"[the CDS] can be defined relative to the full domain, or relative to the sub-domain defined by the coverage of the location probability"*.
        Would these different definitions produce different values for the CDS metric? My understanding is that they would not. If this is the case, can it be made clear that the difference in definition represents only a different way of calculating the same metric.

5.  **§3.1**. Comments / questions on the experiment with meteorological models of different resolutions:
    a.  **Line 250.** *"During the analysis, we noticed that these measurements were highly sensitive to the spatiotemporal resolution of the meteorological data"*.

        This sentence makes it seem as though the resolution of the meteorological data was of passing interest, rather than being a considered factor in the setup of the dispersion model runs. As it is currently presented, I think that the discussion of different meteorological models would fit better in an appendix. To stay in the main body of the paper, I think it requires rephrasing to present the two meteorological models as candidates which are then chosen between based on the comparison with measurements.

    b.  Statistical errors in data analysis can cause meteorological models which are structurally identical (same physics / resolutions) to produce difference analysis data. Please can you discuss / suggest why this is not the primary cause of the differences observed in Fig. 6, and why the difference is produced by structural model differences? Following the suggestion of referee #2, looking at the position of the measurements relative to the plume could help; if the measurements were on the plume edge, they would be sensitive to small meteorological errors (errors in the analysis and short forecasts), so the intrinsic difference between the models might not be significant. Also, reporting the duration of the measurements would help to assess how sensitive they are to short-term changes in the plume.

    c.  Could another researcher get the same met data from MARS if they wanted to repeat this study? What information would they need to do this? (I am not familiar with the MARS archive or FlexExtract, so the answer may be obvious).

6.  **Line 349.** *"the distance metric for the rain water data is rather large, at 1500km"*.

Figure 16 suggests this should say *fallout data* instead (or Fig 16 is incorrectly labelled).

7. **Line 408.** *"source localisation and reconstruction with deposition measurements... can yield useful results in the context of radiological emergency preparedness".*

   I think this part of the sentence needs further justification (I do not question the rest; the work has undoubtedly proved the use of deposition measurements for CTBT-relevant events). What is meant exactly by 'emergency preparedness' and how would the results support it? If 'emergency preparedness' includes 'emergency response' (as mentioned in line 24), how do the results translate into that context, where source term estimates might be required on shorter timescales than used here (i.e. in this application, the first usable deposition measurements were made days(?) after the event, but source term information might be desired within hours in an emergency).

---

## Author Response (AR1)

(Line numbers refer to the author's track-changes file)

**Author's response to referee #1**

**1. The Shershakov et al. (2019) reference on Table 1 is uncompleted. Please, complete or remove this reference from the Table**

Thank you for pointing this out, we have completed the table.

→ Table 1: completed Shershakov et al. (2019) entry.

**2. Line 210: Table 1 shows highly different source terms. Why was the source term from Saunier et al. (2019) chosen? Were the others also tested?**

The Saunier et al. (2019) source term was chosen since it is contains a clearly described temporal release profile. We did not explicitly test other source terms. For the twin experiments, the choice of source term is arbitrary as it is used to generate the synthetic observations.

→ Lines 277-278: added motivation for the choice of source term.

**3. Table 5 wasn´t called anytime in the text. Please, identify each acronym and explore better those scores. What does explain the better correlation for the rain water experiment? Lines 244-245**

Thank you for highlighting this. Non-discussed metrics in the table have been removed and the table is now referenced in the text with a description of each metric.

→ Table 5: removed undiscussed metrics.
→ L312-317: Added references to metrics in Table 5.

**What does explain the better correlation for the rain water experiment?**

We do not know whether the difference is due to the collection method or other sources of error such as the source term or the model itself.

→ L317-318: Provided reasons for poorer correlation.

**4. Line 311: "These releases thus have a small effect on the deposition values". Is the released quantity the only effect? Could the atmospheric conditions like wind speed and direction or atmospheric stability simulated by the model impact this result? Could the first partial release detected by the inversion method related with the wet deposition (Sep, 25th), in truth, refer to the release of the 2 previous days? Could the same release result on significant deposition values for a less favorable dispersion condition?**

In principle this is indeed correct: the deposition is not only proportional to the release quantity, but also the source-receptor-sensitivity which is influenced by the atmospheric transport itself. The source receptor sensitivities for the earlier and main releases are however of similar magnitude. Therefore the release amount, being two orders of magnitude lower, is the dominant factor in the release profile.

→ L400-401: clarified the above.

**5. Line 392: "We found an unexpectedly large impact of the resolution of the meteorological data in Sweden". Why unexpectedly? Since higher resolution modelling can simulate better the physical processes, especially within the Planetary Boundary Layer, the improvement is totally expected. Please, rewrite this phrase.**

We have altered the language used here. However, we would like to clarify that both models are extracted from the same underlying meteorological data, only pre-processed in different resolutions for use with Flexpart. We have now also mentioned this explicitly in Sect. 2.2.

- → L488: removed the word "unexpectedly"
- → L138-147: clarified that all meteorological data used originates from the same ECMWF data. Consequently, slight changes have been made to the wording of already existing parts of this paragraph to improve reading flow.

**Author's response to referee #2**

**1. Line 95: You write that only 30 are left for your analysis. I understand that 35 were discarded, because they were too close to the source, but earlier (line 95), you mentioned that there are 135 measurements in total. What happened to the remaining 70? You write that "… only detections of activity per surface area (i.e. Bq m $^{-2}$ ) were selected." What were the remaining, and why are they not useful?**

The majority of the discarded measurements (65 out of 105) are due to the criteria on the observation window. Applying this criteria and the one for distance leaves five measurements that are in Bq L$^{-1}$. Measurements in Bq m$^{-2}$ are directly suitable for the inverse modelling framework since the deposition fields of Flexpart are output in Bq m$^{-2}$. Due to the low number of measurements in Bq L$^{-1}$, we decided to keep the deposition dataset consistent by using only the Bq m$^{-2}$ values. We have clarified this in the text of the manuscript.

> → Lines 95-98: clarified the above.

**2. Line 95: "We follow the distinction made by Masson et al. (2019) to label 18 of these "activity concentration in rain water" and 12 "dry + wet fallout" … Though the description "rain water" may seem to imply only the collection of wet deposition, in general monitoring networks do not discriminate between dry and wet deposition. It is therefore assumed that both the rain water and fallout measurements contain dry and wet deposition …"**
**I am not sure I understand your approach here. You argue that you cannot justify making a distinction between the two datasets, and you therefore assume that both consist of dry+wet deposition. If this is your assumption, then why do you use the labeling? I find this is a bit confusing.**
**I would probably have preferred one of two options (a) make a distinction between the datasets, and then assume that "rain water" measurements consist of only wet deposition (then use that knowledge for the inversion), or (b) don't make a distinction between the datasets and treat them equally (as you do), but then don't use the labeling and use only the combined dataset and get more robust statistics.**
**If you choose to keep your approach as it is, I think you should further justify why it makes sense to study the two datasets individually. Also, you should make it more clear that "rain water" is only a labeling and not related to the assumed deposition process. In the rest of the article, it is a bit unclear to me how you interpret this label yourself**

Thank you for pointing out this potential confusion. We had initially interpreted the "rain water" measurements to only contain wet deposition, as you suggest. However, after looking at the results and further consultation with experts on deposition measurements, we concluded that these measurements should probably contain dry deposition as well. The reasons for nonetheless keeping the distinction between the two datasets are threefold. Firstly, the distinction is made in the original dataset, so we assume there is an underlying reason for doing so. Secondly, in the inversions and the forward calculation, we find that the "rain water" measurements always contain wet deposition (though some are dominated by dry deposition). In contrast, some "fallout" measurements contain no wet deposition whatsoever. And lastly, we perform and present the inverse modelling for each dataset separately, as well as for the combined datasets. We have clarified the above in our manuscript.

→ We have clarified the above confusion in several places:
L104-105
L321-324
L330-331

**3. Line 100: "The deposition data in the supplementary material of Masson et al. (2019) only provide the start and end dates of the measurements." I miss some information about typical durations of the measurement windows. I think this is a crucial information both for understanding the usefulness of the measurements in the first place, but also for understanding how large the potential error is on the assumed start and end times of the measurements.**

Thank you for pointing out this oversight. We have added an overview of the measurement windows in the manuscript. The duration of the measurement windows range from 1 day (13 measurements) to 7 days (8 measurements), with the rest scattered in between (8 measurements), bar one which is 28 days.

→ L106-108

**4. Line 105: "… the choice to extend the measurement interval by 8 h was made to increase the likelihood of capturing the relevant precipitation event that contributed to any wet deposition." But that goes both ways. You may end up including a precipitation event that was in fact outside of the measurement window.**

We partly agree on this comment. We wanted to ensure that the deposition event was captured for each detection by extending the sampling period . For non-detections, the other approach could be favoured to ensure no deposition takes place but we made a guess that this wouldn't be necessary. The data shows that this was a good guess, as almost all non-detections are properly reproduced (Fig. 5 in new manuscript version).

→ L114-115 & L321-324: clarified the above

**5. Line 150: Can you perhaps elaborate a bit on the Bayesian approach. When you write that you use "… inverse gamma distribution for the combined model and observation uncertainties", do you then mean as prior distribution for the uncertainties, which are assumed unknown? Please, specify this. Further, it would be nice to get a few details about the MCMC sampling method used.**

The Bayesian method uses a Gaussian likelihood where the standard deviation is replaced by an inverse gamma distribution for the combined model and observation uncertainties, which are themselves not estimated. We have now included a more extensive description of the inversion algorithms, with relevant equations in the manuscript. We now also reference the MCMC sampling method that is used for the Bayesian algorithm.

→ L165-194: Added new sections (2.3.1 and 2.3.2) describing the Bayesian and cost function methods in more detail.

**6. Line 220: "A relative measurement error of 50% was chosen." What do you use this for? Is it input to the inversion algorithms? If this is the case, do you assume that this represent only measurement error or combined measurement+modelling error? Finally, if**

**the latter is the case, should you not use a different number for the synthetic vs. real case?**

The relative measurement error referred to $\sigma_{\mathrm{obs}}$ in Eq. (3) of the new manuscript. However, it is actually not used in any of the inversion calculations (i.e. $s_i = \sigma_{\mathrm{mod},i}$), and thus we have removed its mention in the quoted paragraph.

→ L287-294: removed mention of the relative measurement error.

**should you not use a different number for the synthetic vs. real case?**

We agree that this would make sense in principle. However, making the model errors too small will lead to the Bayesian algorithm struggling to converge. Hence, the model errors were not changed for the synthetic case.

**7. Figure 5: Here you show how the modelled dry+wet deposition correspond to the observations of two datasets "rain water" and "fallout". Again, I am not sure I understand the distinction,since you treat the measurements identically. You could increase the amount of data by combining the datasets, and obtain more robust estimates of statistical parameters such as correlation coefficient and fractional bias.**
**One thing that could be interesting is to see how the modelled dry and wet deposition alone correspond to the observations. Can you for example see that the dataset labeled "rain water" seem to be dominated by the modelled wet deposition.**
**If you do not include figures showing this, I would at least appreciate a section somewhere in the text, where you comment on the magnitude of the modelled wet vs. dry deposition values. If you can see that the wet deposition is dominating for the "rain water" dataset, I think this adds to the justification of studying the two datasets individually (cf. my comment nr. 2).**

Here we refer to our response to comment #2.

**8. Line 252: "The higher resolution meteorological data (0.1°, 1h) provides an increase in deposition by one order of magnitude, which is still an underestimation but an improvement over the lower resolution result."**
**I think this is an interesting result, and I especially appreciate the following analysis, where you compare the accumulated column densities for the two different resolutions (Figure 6). However, I notice that both of these measurements are in the dataset labeled "rain water". For gaining further understanding, I think it would be relevant to know if the modelled deposition for these locations is mainly wet, as the labeling suggests (cf. my comment nr 7).**
**Further, did you look at differences in plume structure? Maybe there are some significant differences in the resolved flow? Or perhaps the measurements are taken in an area with large concentration gradients; then even small differences in the in the plume position could explain large discrepancies.**

The Sweden 3 and 6 measurements both consist of around 35% wet and 65% dry deposition. We did indeed look at the plume structure and saw that the difference arises from the fact that the part of the plume that hits Scandinavia during this period is both wider and contains a higher concentration in the high resolution case. We have included a new figure with the resultant spatial deposition patterns (Fig. 7), and an accompanying discussion.

→ L354-360: added discussion and figures comparing the deposition patterns for both sets of meteorological data.

**9. Figure 7: You have not really introduced the term "residual cost" before this figure. Can you perhaps elaborate a bit on the interpretation of this. Do you interpret it as being proportional to a probability density?**

The cost function optimisation is applied to each grid box by fixing the source location while varying the temporal release profile. The end result is a grid of residual costs, where lower cost means a better fit for a release at that location. In this way, the residual cost can be seen as a proxy for the probability density, in the sense that a lower residual cost would correspond to a higher probability. We have included an explanation of this in Sect. 2.3.2 of the manuscript.

→ L190-194: clarified the above.

**10. Figure 9 and 10: It is not until I see these figures that I can guess what type of assumptions you have made about the source term in the two inversion methods. For the cost function based method, you have a release profile with different release rates for each day, while for the Bayesian inversion, it seems that you assume a constant release over a period (described by start time, end time and release rate)? These assumptions should be stated clearly somewherein the text.**

Thank you for pointing this out. We have added a description of the source term parameterisations in Sect. 2.3 for both inversion methods.

→ L177-180

**Since you have decided to use the "twin experiment" with two separate releases as the "truth", it is especially relevant to mention that it is not possible to describe this with the source term discretization you chose for the Bayesian inversion.**
**Further, in Figure 10, I can see the prior distributions you have used for start time, end time and release rate, but I would like to read an explanation somewhere.**

This is indeed correct, we have now stated this explicitly in Sect. 2.3. We have also added a description of the prior distributions.

→ L284-286: mentioned limitations of the Bayesian inference in fitting the separate releases
→ L177-180: added description of prior distributions

**11. Line 328: "Since both dry deposition and air concentration SRS fields are very similar (see Fig. 2), the inverse modelling results are expected to be similar. This is verified with the results as shown." I think this conclusion is very interesting, especially combined with the re-run as described in line 335.**

We thank you for expressing your appreciation.

**12. Line 339 and 400: You conclude that lowering the detection limits of deposition measurement could aid source localization with these measurements. First of all, this statement is probably fair because lowering the detection limit can only improve the results. However, I think you should be careful with concluding too much based on the idealized case with negligible model uncertainties. In "real world" applications, the model**

**uncertainties are expected to be much larger than the measurement uncertainties, so I would not expect to see as big an impact.**

Our conclusion is based on changing the minimal detectable quantities, not the measurement uncertainties. Therefore we do not foresee that larger model uncertainties would invalidate this conclusion. This is also supported by our choice of realistic values for the minimal detectable quantities for the synthetic experiments.

**It would be interesting to see it demonstrated on real data. One option could be to conduct the inversion with all the existing air measurement data (using the real data) and then artificially raise the detection limit of those to see what the impact would be. This is probably a task for a different study, but I think you should at least discuss what impact you would expect when using real measurement data.**

We agree that this would be very interesting, though outside the scope of this study. We have added a discussion in Sect. 3.3: Since the total number of deposition measurements is one order of magnitude smaller - at around 100 - the resulting impact of combining the air concentration and deposition measurements is expected to be minimal since we have shown that a deposition measurement generally contains a similar amount of information as an air concentration measurement for the purposes of source reconstruction.

→ L462-467: added the above-mentioned discussion.

**13. Line 358-360: You describe this problem of the method being over-confident. And while I agree that you would easily be able to point out Mayak in the specific case, it is of course somewhat of a problem. Before I would consider using this method I would appreciate a discussion regarding the cause of the issue as well as possible solutions. Could the problem be that the uncertainties are assumed too small? I am still not 100% sure how the uncertainties are treated in this Bayesian method, because you first mentioned the inverse gamma distribution, and then later wrote that you assume a 50% relative error on the measurements. (cf. my comments 5 and 6). So I look forward to elaborations on this.**

In [1] it was found that the over-confidence can be explained in part by the fact that the Bayesian algorithm prioritises matching smaller detections over larger ones when fitting the source term parameters. [2] evaluates multiple proposals to alleviate this over-confidence. One of the possible avenues is indeed to increase the prescribed model uncertainties.
Nonetheless, this shouldn't necessarily be a problem if the inverse modelling results can be combined with other information such as, for example, a waveform event in the case of a nuclear test or a declaration by a nuclear facility that an accident has occurred.

**14. Figure 15: The Bayesian method gives a very impressive result.**

We appreciate the compliments.

**15. Line 405: "The fallout measurements, however, provide a somewhat worse results for reasons that are unclear." To really discuss this, I still need clarification about why the dataset is split into these two subsets in the first place. If we knew that one dataset is dominated by wet deposition and the other by dry deposition, then I guess that would be the interesting part to discuss. However, if we assume that the two datasets are comparable, then one explanation could be that you have too little data. After all, the**

**"fallout" dataset only consists of only 12 measurements. I hope that your answers to some of my previous questions can also help a bit with the understanding here.**

We agree that limited data might be a cause of the poorer result, and have now mentioned this possibility to the manuscript.

→ L503-504: clarified the above.

[1] P. De Meutter, I. Hoffman, A. W. Delcloo: A baseline for source localisation using the inverse modelling tool FREAR, Journal of Environmental Radioactivity, Volume 273, 2024.

[2] P. De Meutter, I. Hoffman, A. W. Delcloo: A baseline for source localisation using the inverse modelling tool FREAR, HARMO 22 Conference, 2024

**Author's response to referee #3**

**1. Line 21. "Specifically, the release of radionuclides – or more precisely: its accompanying ionizing radiation – can potentially pose …"**
**I think this sentence reads more clearly without the hyphenated expression ("- or more precisely … ionizing radiation –") because the same sentence later states that the danger is from exposure to radiation.**

We have removed the hyphenated expression, as we are in agreement with this comment.

→ Line 21

*2. Line 119. "The retro-plume dispersion was calculated from the start of each measurement".*
**What does this mean? I understand that particles are released at a constant rate over the whole measurement window. If each particle starts being advected as soon as it is released, should this be "from the end of each measurement"?**

Thank you for pointing this out. This should indeed read "from the end of each measurement".

→ L128

**3. §2.4. Comments / questions on the adjoint SRS calculations:**

a. **Fig. 2 is nice, providing an intuitive explanation of the nature of wet / dry deposition observations, and their information content for the purposes of adjoint modelling. It complements the description of the method given in (Eckhardt et al., 2017) (if I am correct that it is the same method?). Since the Eckhardt paper contains many important technical details (e.g. height distribution and masses of model particles), please can you re-reference it in §2.4.**
Fig. 2 is indeed based on the methods from Seibert and Frank (2004) and Eckhardt et al. (2017). We have now referenced these publications in Sect. 2.4 and the figure caption.
→ L249

b. **For completeness, it would be nice for Fig. 2 to include the (adjoint) source functions for the SRS fields, $h(x, y, z, t)$, corresponding to each measurement type (these are proportional to $h$ defined by: $y = \int (h \cdot c) d^3x \, dt$, where $c$ is the activity air concentration field, as in §3 of (Yee et al., 2008)). This essentially rewrites the integrals in a common inner-product framework. Then, I think Fig. 2 would be of even greater use for understanding the SRS method for deposition measurements; in particular, how it relates to the method for air concentration measurements.**
Thank you for this suggestion. We have reformulated Sect. 2.4 and Fig. 2 to include the (adjoint) source functions.
→ L207-258

**4. §2.5. Comments / questions on the performance metrics:**

a. **How is the 'excluded' part of the domain defined in the FDE metric? Is there a threshold for the cost function / Bayesian posterior value? Does the numerical solver / sampler simply fail in these cases?**

The FDE is indeed based on threshold values. A value for 2 is used for the cost function method, and a value of 0 for the Bayesian algorithm. We have now mentioned this in the text.

→ L268

b. **The domain enclosed by the grey rectangle in Fig 1 should be described (e.g. using lat-lon coordinates. Perhaps in the caption to Fig 1?). Then, if others repeat this work but with different computational domains, they will be able to calculate an FDE score which is comparable with the present work.**

The grey rectangle has lower-left corner [40° E, 45° N] and upper-right corner [80° E, 70° N]. We have added this description to both the caption of Fig. 1 and to Sect. 2.3.

→ L269

c. **"[the CDS] can be defined relative to the full domain, or relative to the subdomain defined by the coverage of the location probability".**
**Would these different definitions produce different values for the CDS metric? My understanding is that they would not. If this is the case, can it be made clear that the different in definition represents only a different way of calculating the same metric.**

The different definitions do produce different values for the CDS metric. The CDS for the full domain is related to the subdomain variant by $CDS_{full\ domain} = FDE + (1 - FDE)CDS_{subdomain}$.

**5. §3.1. Comments / questions on the experiment with meteorological models of different resolutions:**

a. **Line 250. "During the analysis, we noticed that these measurements were highly sensitive to the spatiotemporal resolution of the meteorological data".**
**This sentence makes it seem as though the resolution of the meteorological data was of passing interest, rather than being a considered factor in the setup of the dispersion model runs. As it is currently presented, I think that the discussion of different meteorological models would fit better in an appendix. To stay in the main body of the paper, I think it requires rephrasing to present the two meteorological models as candidates which are then chosen between based on the comparison with measurements.**

We have changed the wording in both Sect. 2.3 and 3.1 to clarify that we evaluate the meteorological data as a part of the model setup evaluation.

→ L138-139
→ L331-332

b. **Statistical errors in data analysis can cause meteorological models which are structurally identical (same physics / resolutions) to produce difference analysis data. Please can you discuss / suggest why this is not the primary cause of the differences observed in Fig. 6, and why the difference is produced by structural model differences? Following the suggestion of referee #2, looking at the position**

**of the measurements relative to the plume could help; if the measurements were on the plume edge, they would be sensitive to small meteorological errors (errors in the analysis and short forecasts), so the intrinsic difference between the models might not be significant. Also, reporting the duration of the measurements would help to assess how sensitive they are to short-term changes in the plume.**
We observe a significant difference in the plume structure itself, where the bifurcated part of the plume that hits Scandinavia is much wider and contains higher concentrations in the high resolution versus the low resolution met data. We have added this information, and an accompanying figure of the resultant deposition fields to Sect. 3.1.
→ L354-360

c. **Could another researcher get the same met data from MARS if they wanted to repeat this study? What information would they need to do this? (I am not familiar with the MARS archive or FlexExtract, so the answer may be obvious).**
The details to access the MARS archive can be found on https://www.ecmwf.int/en/forecasts/access-forecasts/access-archive-datasets. Then one can use FlexExtract on the met data to extract it for use in Flexpart, in the desired resolution.

**6. Line 349. "the distance metric for the rain water data is rather large, at 1500km".Figure 16 suggests this should say fallout data instead (or Fig 16 is incorrectly labelled).**

Thank you for pointing out this error, it should indeed read "fallout data".

→ L439

**7. Line 408. "source localisation and reconstruction with deposition measurements… can yield useful results in the context of radiological emergency preparedness".**
**I think this part of the sentence needs further justification (I do not question the rest; the work has undoubtedly proved the use of deposition measurements for CTBT-relevant events). What is meant exactly by 'emergency preparedness' and how would the results support it? If 'emergency preparedness' includes 'emergency response' (as mentioned in line 24), how do the results translate into that context, where source term estimates might be required on shorter timescales than used here (i.e. in this application, the first usable deposition measurements were made days(?) after the event, but source term information might be desired within hours in an emergency).**

The context we refer to is both radiological emergency preparedness and response. We have now made this consistent in the manuscript. Emergency preparedness encapsulates the preparation one requires to properly respond to an unforeseen radiological emergency situation. The results of our study are useful for both emergency preparedness and response since we show that having the ability to perform deposition measurements can aid in source reconstruction, and hence inform the further response.
In the Ru-106 case, the deposition measurements were made several days after the event since that is the time it took the plume to reach these locations. However, it is also true that in general there will be a larger time delay to obtain deposition measurements compared to air concentration measurements, the latter of which are routinely measured in detector networks while deposition measurements are not. Nonetheless, deposition measurements can be made in

an emergency response scenario. As part of emergency preparedness there (can) exist mobile units which are equipped with deposition measurement apparatus. Depending on detection limits, measurements can be made within the order of hours to aid real-time source reconstruction. Furthermore, it would also be theoretically possible to build automatic deposition detectors in detector networks.

Deposition measurements can also be used to retrospectively estimate the source term, as has been done after the Fukushima accident. This information can be used to implement emergency measures based on the estimated impact on the food chain and total dose exposure.

- → L24
- → L507